# Control of protein synthesis and memory by GluN3A-NMDA receptors through inhibition of GIT1/mTORC1 assembly

María J Conde-Dusman[1,2,3†], Partha N Dey[2,4†], Óscar Elía-Zudaire[1†], Luis G Rabaneda[1,2,5], Carmen García-Lira[1], Teddy Grand[6], Victor Briz[7], Eric R Velasco[8], Raül Andero[9,10,11], Sergio Niñerola[1], Angel Barco[1], Pierre Paoletti[6], John F Wesseling[1], Fabrizio Gardoni[12], Steven J Tavalin[13], Isabel Perez-Otaño[1,2*†]

[1]Instituto de Neurociencias (UMH-CSIC), Alicante, Spain; [2]Centro de Investigación Médica Aplicada (CIMA), University of Navarra, Pamplona, Spain; [3]Centre for Developmental Neurobiology, Institute of Psychiatry, King's College London, London, United Kingdom; [4]National Eye Institute, National Institutes of Health, Bethesda, United States; [5]Institute of Science and Technology Austria, Klosterneuburg, Austria; [6]Institut de Biologie de l'Ecole Normale Supérieure/CNRS/INSERM, Paris, France; [7]Centro de Biología Molecular Severo Ochoa (UAM-CSIC), Madrid, Spain; [8]Institut de Neurociències, Universitat Autònoma de Barcelona, Bellaterra, Spain; [9]Institut de Neurociències, Departament de Psicobiologia i de Metodologia de les Ciències de la Salut, Unitat de Neurociència Traslacional, Parc Taulí Hospital Universitari, Institut d'Investigació i Innovació Parc Taulí (I3PT), Universitat Autònoma de Barcelona, Bellaterra, Spain; [10]Centro de Investigación Biomédica en Red de Salud Mental (CIBERSAM), Instituto de Salud Carlos III, Madrid, Spain; [11]ICREA, Barcelona, Spain; [12]Department of Pharmacological and Biomolecular Sciences, University of Milan, Milan, Italy; [13]Department of Pharmacology, Addiction Science, and Toxicology, University of Tennessee Health Science Center, Memphis, United States

*For correspondence:
otano@umh.es

†These authors contributed equally to this work

Competing interest: The authors declare that no competing interests exist.

**Abstract** De novo protein synthesis is required for synapse modifications underlying stable memory encoding. Yet neurons are highly compartmentalized cells and how protein synthesis can be regulated at the synapse level is unknown. Here, we characterize neuronal signaling complexes formed by the postsynaptic scaffold GIT1, the mechanistic target of rapamycin (mTOR) kinase, and Raptor that couple synaptic stimuli to mTOR-dependent protein synthesis; and identify NMDA receptors containing GluN3A subunits as key negative regulators of GIT1 binding to mTOR. Disruption of GIT1/mTOR complexes by enhancing GluN3A expression or silencing GIT1 inhibits synaptic mTOR activation and restricts the mTOR-dependent translation of specific activity-regulated mRNAs. Conversely, GluN3A removal enables complex formation, potentiates mTOR-dependent protein synthesis, and facilitates the consolidation of associative and spatial memories in mice. The memory enhancement becomes evident with light or spaced training, can be achieved by selectively deleting GluN3A from excitatory neurons during adulthood, and does not compromise other aspects of cognition such as memory flexibility or extinction. Our findings provide mechanistic insight into synaptic translational control and reveal a potentially selective target for cognitive enhancement.

## Introduction

Memories are thought to be encoded through formation and modification of the synaptic connections between neurons. Lasting memory encoding requires de novo mRNA and protein synthesis in response to neuronal activity and sensory experience. It entails the transcription of immediate-early genes (IEGs) to mRNA, and the protein products of some IEG transcripts mediate structural and functional modifications of synapses (*Yap and Greenberg, 2018*). However, transcription occurs in the cell body and generates a neuron-wide pool of mRNAs, whereas only a fraction of synapses of any individual neuron are modified by a given memory (*Holtmaat and Caroni, 2016*; *Josselyn and Tonegawa, 2020*). To ensure input specificity, transcription is coupled to local mechanisms that restrict the effects of activity-induced gene products to selected synapses (*Wang et al., 2010*).

One of these mechanisms is thought to be the local, synapse-specific translation of mRNA into protein (*Holt et al., 2019*; *Klann and Dever, 2004*; *Sossin and Costa-Mattioli, 2019*). The main rate-limiting step in translation is initiation, which is regulated by the phosphorylation of two separate proteins: the eukaryotic initiation factor 2α (eIF2α) and the mTOR ('mechanistic target of rapamycin') serine/threonine kinase. Manipulations of eIF2α phosphorylation have been implicated in synapse plasticity and memory (*Costa-Mattioli et al., 2007*; *Sharma et al., 2020*; *Shrestha et al., 2020b*), but evidence for a role in local translation is lacking. mTOR could in principle afford more selective translational control. mTOR forms at least two distinct multiprotein complexes, mTORC1 and mTORC2. mTORC1 is defined by the presence of Raptor, an adaptor protein which recruits mTOR substrates to promote the translation of specific mRNAs, and compartmentalized activation has been shown to be essential for mTORC1 responses to nutrients in nonneuronal cells (*Liu and Sabatini, 2020*). In neurons, components of mTORC1 localize to axons, dendrites, and synapses (*Poulopoulos et al., 2019*; *Takei et al., 2004*; *Tang et al., 2002*), and pharmacological inhibition of mTORC1 with rapamycin blocks long-lasting synaptic plasticity and memory formation (*Cammalleri et al., 2003*; *Hou and Klann, 2004*; *Stoica et al., 2011*; *Tang et al., 2002*). Moreover, dysregulated translation is a feature in diseases of cognition, from autism to intellectual disability, and many of the mutations associated with these diseases affect genes encoding negative regulators of mTORC1 (*Costa-Mattioli and Monteggia, 2013*; *Lipton and Sahin, 2014*). However, it is currently unclear how mTOR activation might be controlled at specific synapses and linked to mechanisms that gate learning and memory.

The most intensively studied mechanism gating learning and memory involves the NMDA-type glutamate receptor (NMDAR). NMDARs contain multiple subunits, including an obligatory GluN1 subunit, various GluN2 (A–D) and, for some subtypes, one of the GluN3 (A–B) subunits (*Paoletti et al., 2013*). Conventional subtypes containing GluN1 and GluN2 trigger gene expression programs that mediate the strengthening and stabilization of active synapses and the persistent storage of information (*Lyons and West, 2011*). By contrast, nonconventional subtypes containing the GluN3A subunit (GluN3A-NMDARs) inhibit many of these synaptic modifications (*Pérez-Otaño et al., 2016*). Synapses that express GluN3A are resistant to the induction of long-lasting functional and structural plasticity, and memories fade more quickly in mutant mice with enhanced GluN3A expression (*Kehoe et al., 2014*; *Roberts et al., 2009*). In line with this work in mice, human genetic studies correlate enhanced cognitive performance with low GluN3A levels or variations in *GRIN3A* (human gene encoding GluN3A) (*Gallinat et al., 2007*; *Papenberg et al., 2014*; *Sadat-Shirazi et al., 2019*); and GluN3A dysregulation in humans is linked to cognitive impairment in schizophrenia (*Greenwood et al., 2019*; *Mueller and Meador-Woodruff, 2004*; *Ohi et al., 2015*; *Takata et al., 2013*), Huntington's disease (*Marco et al., 2013*; *Marco et al., 2018*), addiction, and other pathologies (*Huang et al., 2017*; *Pérez-Otaño et al., 2016*; *Sarker et al., 2019*; *Yang et al., 2015*; *Yuan et al., 2013*). We reasoned that understanding the underlying mechanisms would yield insight into the brain processes that constrain long-term memory formation and might uncover targets for therapeutic intervention.

Here, we report that GluN3A-NMDARs selectively and negatively regulate synaptic mTORC1-dependent translation without affecting neuron-wide transcriptional activation. The negative regulation is mediated by inhibition of the assembly of mTOR complexes that contain the postsynaptic adaptor GIT1 (G-protein-coupled receptor kinase-interacting protein) and Raptor. GIT1/mTORC1 complexes are located at or near synaptic sites, and couple mTORC1 kinase activity to synaptic stimulation. Through biochemical, mouse genetics, and behavioral approaches, we further show that GluN3A deletion increases the availability of GIT1/mTORC1 complexes, boosts mTORC1-dependent protein synthesis, and facilitates long-term memory formation. The advantage is selectively evident

when mice are subjected to weak training behavioral paradigms; can be reversed by the mTORC1 inhibitor rapamycin; and unlike the memory enhancement seen after manipulations of general translational regulators, is not associated with deficits in memory flexibility or extinction (*Shrestha et al., 2020a*). Our findings identify a novel regulatory mechanism whereby GluN3A/GIT1 interactions set local modes of protein synthesis and gate memory formation, and reveal a potentially selective target for correcting cognitive impairment in pathological contexts.

## Results

### Selective inhibition of activity-dependent gene expression by GluN3A at the post-transcriptional level

GluN3A expression is pervasive during postnatal brain development, and regulated removal allows for the activity-dependent stabilization or elimination of excess synapses (*Pérez-Otaño et al., 2016*). To assess whether GluN3A-NMDARs modulate activity-dependent gene expression, we expressed GluN3A in cultured cortical neurons over the stage when endogenous downregulation normally occurs (days in vitro [DIV] 9–14, ~ postnatal days P8–P16 in vivo, *Figure 1A*; *Figure 1—figure supplement 1*; *Kehoe et al., 2014*). We used lentiviral vectors where expression is targeted to neurons by the synapsin 1 promoter and induced synaptic activity with bicuculline, which inhibits γ-aminobutyric acid (GABA) transmission and triggers bursts of action potential firing (*Hardingham et al., 2002*).

As expected, bicuculline induced a robust expression of IEGs implicated in the consolidation of synaptic modifications and memories, including *Arc, Fos*, and *Zif268/Egr1* (*Flavell and Greenberg, 2008*; *Figure 1B*). Enhancing GluN3A expression largely reduced the induction of Arc and Fos proteins while Zif268 induction was unaffected, indicating that GluN3A selectively inhibits specific activity-dependent signaling pathways (*Figure 1B*). Analysis at the mRNA level demonstrated that modulation occurs downstream of gene transcription: *Arc*, *Fos*, and *Zif268* mRNA levels were strongly induced by bicuculline in both control and GluN3A-infected neurons, and no differences were observed in the time-courses or magnitude of induction (*Figure 1C*). Unchanged transcription was in-line with intact activation of the phosphorylation of extracellular signal-regulated kinase (ERK1/2) and CREB (*Figure 1—figure supplement 1B*), the two major pathways for activity-dependent transcription (*Flavell and Greenberg, 2008*). By contrast, the general NMDAR antagonist D-2-amino-5-phosphonovaleric acid (APV) inhibited all signaling pathways analyzed and the induction of IEGs at both mRNA and protein levels (*Figure 1—figure supplement 1C*, D).

An analogous dissociation between protein and transcript levels of a subset of IEGs was observed when GluN3A-infected neurons were stimulated with the neurotrophin BDNF (*Figure 1—figure supplement 1E*, F), a potent inducer of gene expression at both transcriptional and translational levels (*Rao et al., 2006*). Whole transcriptome RNAseq analyses confirmed that transcriptional responses to bicuculline or BDNF were unaffected by GluN3A expression (*Figure 1D*; *Figure 1—figure supplement 2*). Together these results indicated that GluN3A-NMDARs repress the translation of specific activity-regulated mRNAs without affecting global transcriptional programs of gene expression. Inhibited induction of IEGs by GluN3A was not rescued by pretreatment with the proteasome inhibitor MG-132 (*Figure 1E*), ruling out alternative mechanisms such as enhanced proteasome-dependent degradation (*Rao et al., 2006*).

### GluN3A inhibits mTORC1-dependent translation of IEGs

We thus turned to protein synthesis pathways to search for mechanisms underlying the selective inhibition of gene expression by GluN3A. We focused on mTORC1 because it has been shown to couple synaptic signals including BDNF and NMDAR activation to translation of specific mRNAs in dendrites and synapses (*Takei et al., 2004*; *Tang et al., 2002*). mTORC1 signaling was strongly activated by bicuculline in DIV14 cortical neurons, as shown by phosphorylation of mTOR on Ser$^{2448}$ (a reliable readout of mTORC1 kinase activity; see *Chiang and Abraham, 2005*) and of its downstream effectors, the p70-kDa ribosomal protein S6 kinase (S6K, Thr$^{389}$) and the ribosomal protein S6 (Ser$^{240-4}$, *Figure 2A and B*). The effects were blocked by APV and the NMDAR open-channel blocker MK-801, confirming NMDAR dependence in our model (*Figure 2—figure supplement 1*).

The phosphorylation of mTOR, S6K, and S6 following bicuculline treatment was significantly reduced in GluN3A-infected neurons, indicating that GluN3A interferes with synaptic mTORC1 activation

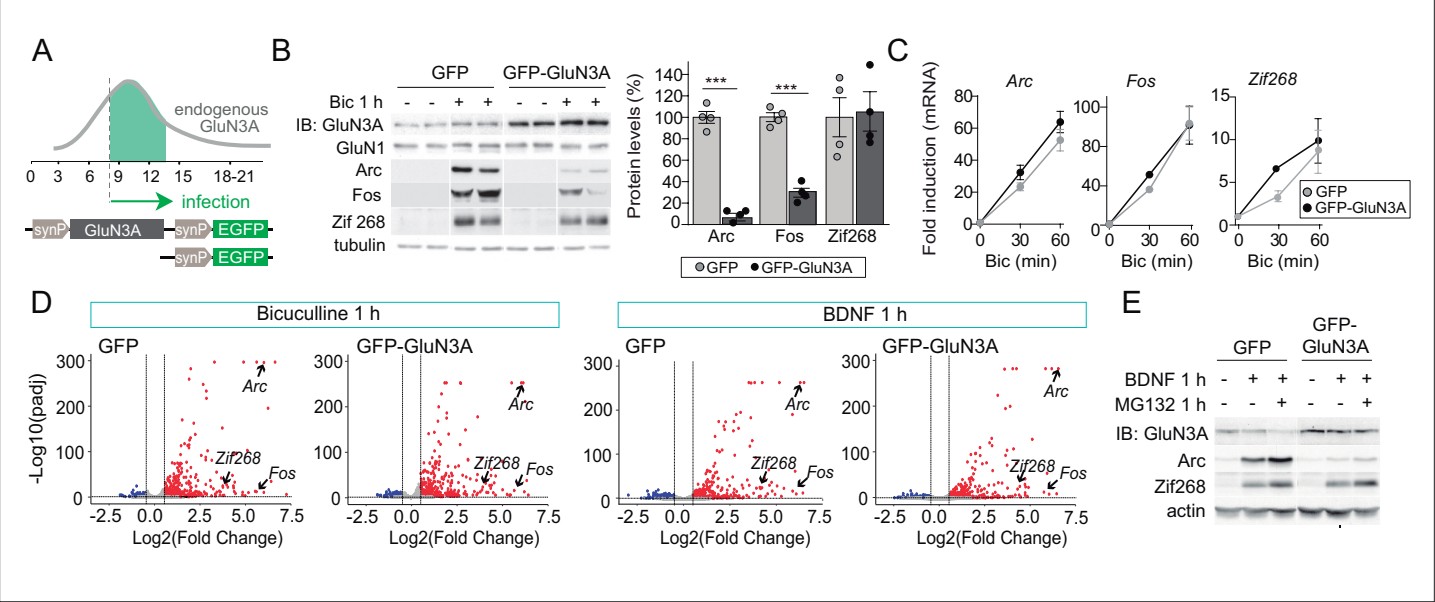

**Figure 1.** GluN3A inhibits the activity-dependent induction of a subset of immediate-early genes (IEGs). (**A**) Timeline of endogenous GluN3A expression and downregulation and of lentiviral infections. Rat cortical neurons in primary culture were infected on days in vitro (DIV) 9 with lentiviruses where Green fluorescent protein (GFP) or GluN3A and GFP (GFP-GluN3A) expression is driven by the human synapsin 1 promoter (synP). (**B, C**) DIV14 neurons were treated with bicuculline (50 μM, 1 hr) and matching samples collected for immunoblot and mRNA analyses (n = 4 from two independent cultures; ***p < 0.001, two-tailed unpaired t-test). (**B**) Left, representative western blots show that GluN3A inhibits the induction of the IEGs Arc and Fos but not Zif268. Right, signal intensities of indicated proteins as percentage of stimulated GFP-infected neurons. (**C**) quantitative Reverse Transcription-Polymerase Chain Reaction (qRT-PCR) analysis of IEG mRNA induction. Plotted values are fold-induction relative to non-stimulated neurons. (**D**) Volcano plots presenting the RNAseq-based differential expression analysis in DIV14 neurons treated with bicuculline or Brain-derived neurotrophic factor (BDNF) for 1 hr (n = 2–4 from two independent cultures). (**E**) DIV14 neurons were treated with MG132 (30 μM). A representative western blot probed with the indicated antibodies is shown. In immunoblot analyses, tubulin or actin was used as a loading control and GluN1 as a measure of potential effects of GluN3A on overall NMDAR numbers. Histograms in this and subsequent figures are mean ± standard error of the mean (SEM).

The online version of this article includes the following figure supplement(s) for figure 1:

**Source data 1.** Western blots for immediate-early gene (IEG) induction in GFP and GFP-GluN3A-infected neurons after bicuculline treatment.

**Source data 2.** Western blots for bicuculline induction of immediate-early genes (IEGs) in GFP and GFP-GluN3A-infected neurons in the presence of MG132.

**Figure supplement 1.** Selective versus global effects of GluN3A expression and general NMDAR blockade on activity-dependent signaling.

**Figure supplement 1—source data 1.** Annotated western blots and original scans.

**Figure supplement 1—source data 2.** Annotated western blots and original scans.

**Figure supplement 1—source data 3.** Annotated western blots and original scans.

**Figure supplement 1—source data 4.** Annotated western blots and original scans.

**Figure supplement 2.** RNAseq analysis of activity-dependent gene expression.

(**Figure 2B**). Two experiments linked mTORC1 inhibition to the altered production of IEGs. First, low concentrations of rapamycin (100 nM) that inhibit mTORC1 but not mTORC2 (**Zhu et al., 2018**), blocked Arc and Fos translation in response to bicuculline without affecting Zif268, demonstrating selective mTORC1 dependence (**Figure 2C**). By contrast, the general protein synthesis inhibitor anisomycin fully suppressed Arc, Fos, and Zif268 induction (**Figure 2—figure supplement 1B**). Second, restoring mTORC1 signaling in GluN3A-infected neurons by expressing a constitutively active form of Rheb, the main upstream activator of mTORC1, was sufficient to normalize IEG induction (**Figure 2D**).

Conversely, lentiviral knockdown of GluN3A in cortical neurons with a validated short hairpin RNA (**Kehoe et al., 2014**) enhanced mTORC1 activity (**Figure 3A and B**) and potentiated the induction of Arc and Fos by bicuculline or BDNF (**Figure 3C and D**). Increased phosphorylation of S6K and S6 was additionally detected in hippocampal lysates from mice lacking GluN3A (Grin3a−/−) relative to wild-type littermates (**Figure 3E**), confirming a role of GluN3A in limiting mTORC1 signaling in vivo.

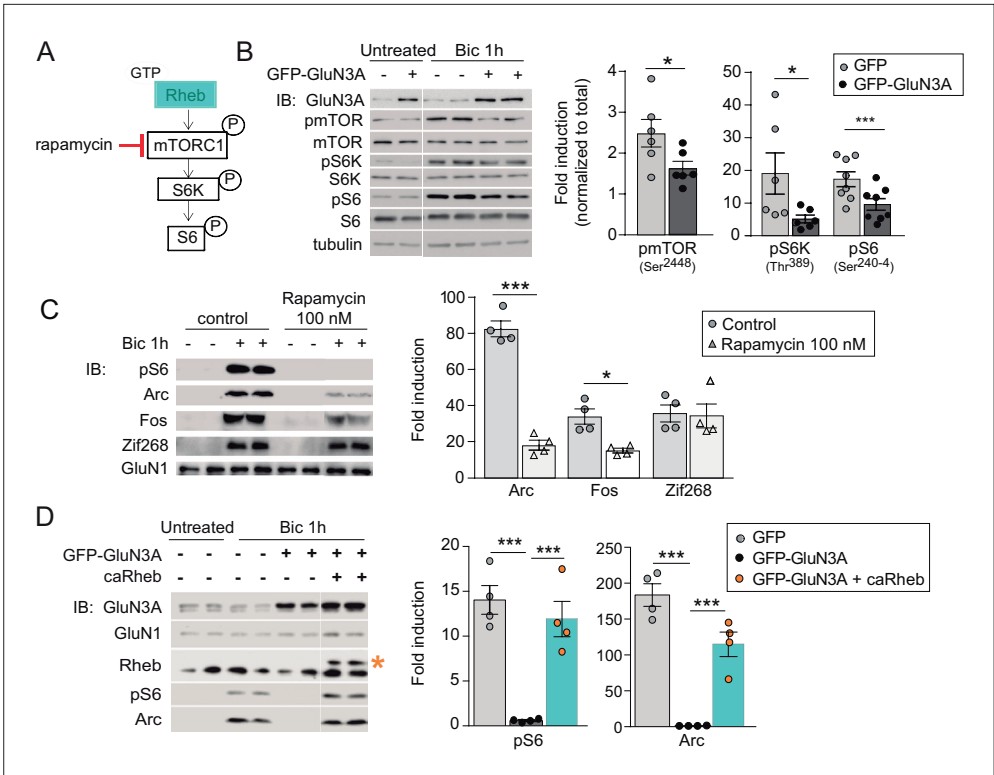

**Figure 2.** GluN3A inhibits the activation of mTORC1 signaling by synaptic stimuli. (**A**) Schematic of the mTORC1 signaling pathway. (**B**) Left, representative western blots of primary rat cortical neurons infected with GFP and GFP-GluN3A (days in vitro [DIV] 9) and treated with bicuculline at DIV14. Right, fold-induction of phosphorylated mTOR, S6 kinase (S6K), and S6 normalized to total protein ($n$ = 6–8 from three to four independent cultures; *p < 0.05, ***p < 0.001, two-tailed paired $t$-test). (**C**) mTOR is required for activity-dependent induction of Arc and Fos but not Zif268. Left, representative western blots of DIV14 neurons stimulated with bicuculline after preincubation with rapamycin (100 nM, 1 hr before and during bicuculline treatment). Right, fold-induction of immediate-early genes (IEGs) in response to bicuculline ($n$ = 4 from two independent cultures; *p < 0.05, ***p < 0.001, two-tailed paired $t$-test). (**D**) Reactivation of mTOR in GFP-GluN3A-infected neurons by adeno-associated virus (AAV) driven constitutively active Rheb (caRheb) rescues Arc induction. Left, representative western blots of neurons infected with lentiviral GFP-GluN3A and AAV-caRheb and treated with bicuculline. Right, fold-induction by bicuculline of pS6 and Arc in the indicated conditions ($n$ = 4 from two independent cultures; ***p < 0.001, one-way analysis of variance [ANOVA] followed by Tukey's test).

The online version of this article includes the following figure supplement(s) for figure 2:

**Source data 1.** Western blots for mTOR and downstream effector phosphorylation in GFP and GFP-GluN3A-infected cortical neurons after bicuculline treatment.

**Source data 2.** Western blots for rapamycin dependence of immediate-early gene (IEG) induction in DIV14 cortical neurons by bicuculline treatment.

**Source data 3.** Western blots for S6 phosphorylation and Arc induction by bicuculline in GFP and GFP-GluN3A-infected cortical neurons in the presence of caRheb.

**Figure supplement 1.** General inhibition of the activity induction of immediate-early genes (IEGs) by anisomycin.

**Figure supplement 1—source data 1.** Annotated western blots and original scans.

**Figure supplement 1—source data 2.** Annotated western blots and original scan.

## mTORC1 inhibition requires GluN3A C-terminal domain interactions

GluN3A subunits confer unique biophysical properties to NMDARs, including reduced channel conductance and calcium permeability, and enable distinct interactions with signaling/scaffolding proteins via their intracellular C-terminal tail (*Pérez-Otaño et al., 2016*). To dissect their contribution to inhibited mTORC1 signaling, we derived primary cortical neurons from *Grin3a*−/− mice and reexpressed full-length GluN3A, a mutant where the distal 33 amino acids of the GluN3A C-terminus

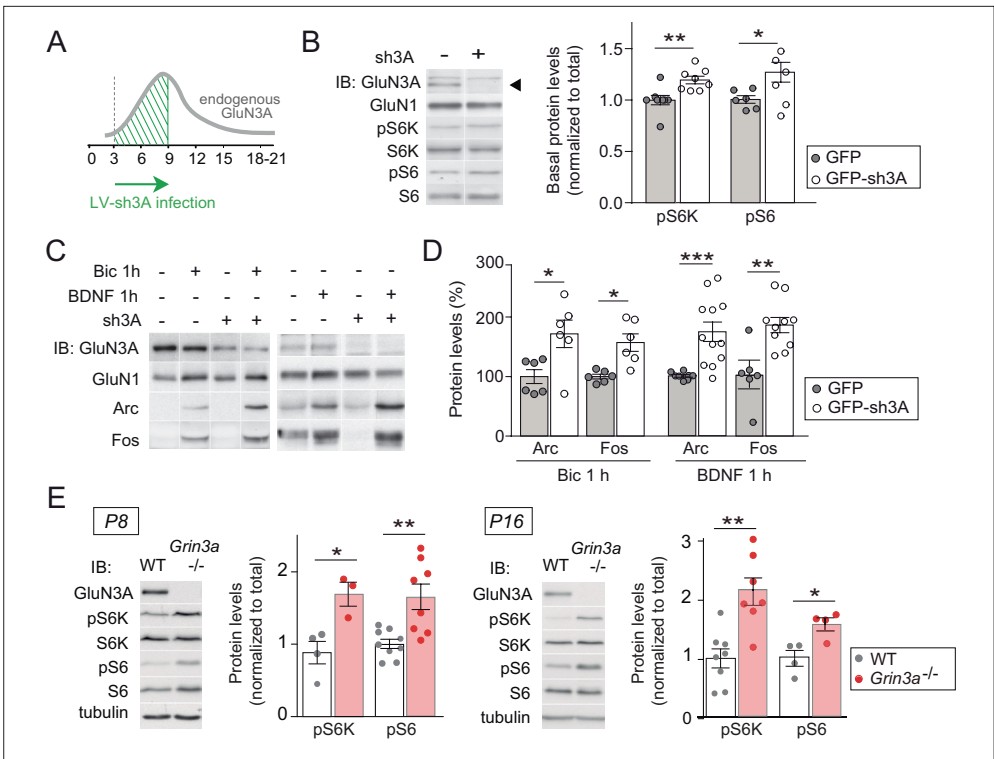

**Figure 3.** GluN3A deletion potentiates synaptic mTORC1 signaling. (**A**) Primary rat cortical neurons were infected on days in vitro (DIV) 3 with lentiviruses expressing GFP alone or along with a small hairpin RNA (shRNA) against GluN3A (GFP-sh3A) and collected at DIV7–9, when GluN3A expression is maximal. (**B**) Representative blots and quantification of phosphorylated S6 kinase (S6K) and S6 normalized to total protein ($n$ = 6–8 from three to four independent cultures; *$p < 0.05$, **$p < 0.01$ two-tailed paired $t$-test). Arrow marks specific GluN3A band. (**C, D**) Representative western blots and quantification of immediate-early gene (IEG) induction by bicuculline or BDNF ($n$ = 6–12 from three independent cultures; *$p < 0.05$, **$p < 0.01$, ***$p < 0.001$, two-tailed paired $t$-test). Data plotted as percentage of stimulated control GFP-infected neurons. (**E**) Immunoblots and quantification of S6K and S6 phosphorylation in lysates from wild-type (WT) and $Grin3a–/–$ hippocampi ($n$ = 4–8 mice; *$p < 0.05$, **$p < 0.01$, two-tailed unpaired $t$-test).

The online version of this article includes the following figure supplement(s) for figure 3:

**Source data 1.** Western blots for S6 kinase (S6K) and S6 phosphorylation in control and sh3A-infected days in vitro (DIV) 7 cortical neurons.

**Source data 2.** Western blots for Arc and Fos induction by bicuculline and BDNF in control and sh3A-infected days in vitro (DIV) 7 cortical neurons.

**Source data 3.** Western blots for S6 kinase (S6K) and S6 phosphorylation in lysates from P8 and P16 wild-type and $Grin3a–/–$ hippocampi.

have been deleted and lacks synapse destabilizing activity (GluN3A1082Δ) (*Fiuza et al., 2013*; *Kehoe et al., 2014*), or GFP as a control (*Figure 4A*). While full-length GluN3A rescued the enhanced mTOR activation and hyperinduction of Arc and Fos proteins by bicuculline or BDNF in $Grin3a–/–$ cultures, the GluN3A1082Δ mutant failed to do so (*Figure 4B–E*). Neither GluN3A nor GluN3A1082Δ modified the activation of other signaling pathways such as ERK1/2 phosphorylation or the induction of Zif268 in $Grin3a–/–$ neurons (*Figure 4D*).

Since GluN3A and GluN3A1082Δ display similar distributions and cell surface targeting (*Fiuza et al., 2013*), the differences we observed are unlikely to stem from altered subcellular localization. We evaluated whether the deletion alters ion fluxes via GluN3A-NMDARs by analyzing electrophysiological responses to glutamate of GluN3A and GluN3A1082Δ when coexpressed with GluN1 and GluN2A in HEK293 cells. The relative calcium permeability was estimated by measuring the shift in reversal potential ($\Delta E_{rev}$) of recombinant NMDAR currents induced by changing extracellular $Ca^{2+}$ (*Perez-Otano et al., 2001*). GluN3A and GluN3A1082Δ yielded similarly reduced shifts in $\Delta E_{rev}$ relative

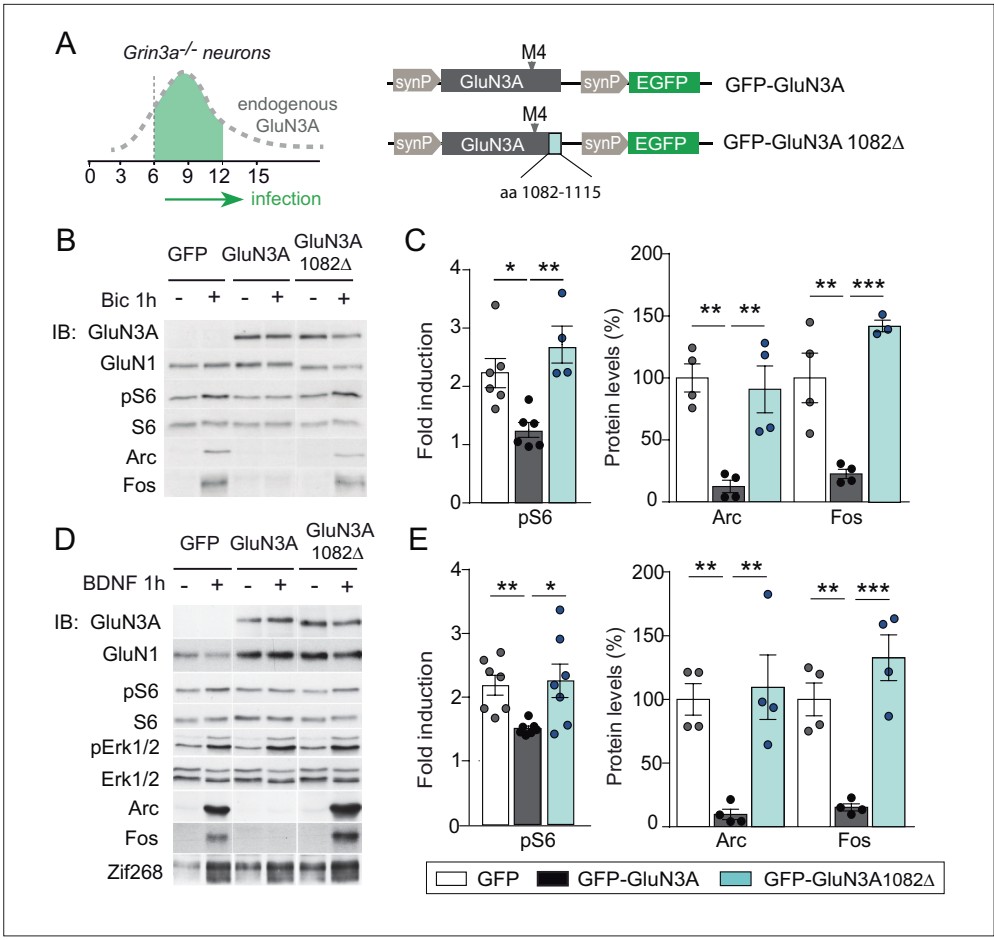

**Figure 4.** mTORC1 inhibition is mediated by GluN3A C-terminal domain interactions. (**A**) Cortical neurons from *Grin3a−/−* mice were infected on days in vitro (DIV) 6 with lentiviruses expressing GFP, GFP-GluN3A, or GFP-GluN3A1082Δ, and stimulated with bicuculline or BDNF at DIV12. (**B, D**) Representative western blots of lysates from bicuculline or BDNF-treated neurons probed for the indicated antibodies. (**C, E**) Induction of phosphorylated S6 (normalized to total levels), Arc and Fos by bicuculline or BDNF (*n* = 3–7 from two to four independent cultures, *p < 0.05, **p < 0.01, ***p < 0.001 analysis of variance [ANOVA] followed by Tukey's test).

The online version of this article includes the following figure supplement(s) for figure 4:

**Source data 1.** Western blots for mechanistic target of rapamycin (mTOR) effector phosphorylation and Arc and Fos induction in GFP, GFP-GluN3A, and GFP-GluN3A1082Δ-infected cortical neurons after bicuculline treatment.

**Source data 2.** Western blots for mechanistic target of rapamycin (mTOR) effector phosphorylation and Arc and Fos induction in GFP, GFP-GluN3A, and GFP-GluN3A1082Δ-infected cortical neurons after BDNF treatment.

**Figure supplement 1.** Electrophysiological properties of recombinant NMDA and excitatory glycine receptors containing full-length or truncated GluN3A.

to conventional GluN1/GluN2A NMDARs, confirming that the mutant incorporated into functional triheteromeric GluN3A-NMDARs and arguing against differences in $Ca^{2+}$ permeability (***Figure 4—figure supplement 1***). In addition, both GluN3A versions drove comparable reductions in current densities relative to conventional NMDARs (***Figure 4—figure supplement 1B***). Along with nonconventional NMDARs, GluN3A subunits can form glycine-gated diheteromeric GluN1/GluN3 receptors (***Pérez-Otaño et al., 2016***). Thus, we additionally examined whether the deletion modified responses to glycine of GluN1/GluN3 receptors taking advantage of the CGP-78608 compound (***Grand et al., 2018***), but no differences were found (***Figure 4—figure supplement 1C***). The absence of ionotropic differences favored the hypothesis that the inhibition of mTOR signaling requires metabotropic interactions of GluN3A-NMDARs, possibly modulating its association with synaptic adaptors or scaffolds.

## GluN3A expression modulates the assembly of synaptic GIT1/mTORC1 complexes

A leading candidate is the multifunctional adaptor GIT1. GIT1 is enriched in postsynaptic compartments and binds the 33 amino acids of the GluN3A C-terminus that we show above are required for mTORC1 inhibition (*Fiuza et al., 2013*). Although best known for its role in actin signaling (*Zhang et al., 2003*), GIT1 has been detected in mTOR immunoprecipates from mouse astrocytes by mass spectrometry (*Smithson and Gutmann, 2016*) though a function for this association could not be established. Using reciprocal immunoprecipitation with GIT1 and mTOR antibodies, we isolated GIT1/mTOR complexes from lysates of microdissected hippocampal (*Figure 5A*) and cortical (not shown) tissue. We chose detergent conditions that preserve mTOR interactions with Raptor and Rictor (0.3 % CHAPS) to further characterize the composition of the complex, and were able to identify Raptor (but not the mTORC2 component Rictor) in GIT1 immunoprecipates. The mTOR antibody pulled down both, validating our assay conditions (*Figure 5A*). The GIT1-binding protein and Rac1 activator βPIX was also pulled down by the mTOR antibody while the control presynaptic protein synaptophysin was not (*Figure 5A*). We additionally detected phosphorylated mTOR at Ser$^{2448}$ in GIT1 immunoprecipitates, demonstrating GIT1/mTORC1 complex functionality (*Figure 5B*).

We then examined the subcellular localization of GIT1/mTORC1 complexes using in situ proximity ligation assay (PLA) with antibodies against GIT1 and mTOR. PLA puncta were present along dendritic shafts often localized within or at the base of dendritic spines (*Figure 5C*), suggesting that GIT1 positions mTORC1 near synaptic sites to mediate dendritic translation in response to synaptic signals. To test this, we stimulated cortical neurons with bicuculline or BDNF and quantified mTOR phosphorylation in total lysates and GIT1 immunoprecipitates. Both bicuculline and BDNF induced large increases in the phosphorylation of GIT1-bound mTOR on Ser$^{2448}$ (*Figure 5D and E*). Importantly, the phosphorylation of GIT1-bound mTOR was much higher than phosphorylation of the total cellular mTOR pool (BDNF: 1.98 ± 0.38- fold increase in total lysates vs. 4.2 ± 1.15 in GIT1 immunoprecipates; bicuculline: 1.42 ± 0.15 vs. 4.63 ± 1.24), consistent with a role for GIT1 in nucleating synaptic mTORC1 activation. Further evidence came from GIT1 loss-of-function experiments. Lentiviral knockdown of GIT1 blunted the activation of mTORC1 by BDNF, as shown by reduced phosphorylation of S6 and S6K (*Figure 5F and G*), and inhibited mTORC1-dependent protein synthesis assessed using a nonradioactive puromycin-labeling assay (SUnSET) (*Figure 5H*). Arc translation was also reduced, as judged by loss of rapamycin sensitivity relative to control neurons, while Zif268 which is mTORC1 independent was unaffected (*Figure 5F and G*). Collectively, these experiments demonstrated the existence of mTORC1 complexes composed of GIT1, mTOR, and Raptor that mediate mTORC1 signaling in response to synaptic stimuli.

## GluN3A/GIT1 interactions control the emergence of mTORC1-dependent protein synthesis

We further found that the abundance of GIT1/mTORC1 complexes is regulated throughout postnatal development. GIT1/mTORC1 complexes were readily observed in P16, but not P7 or P10, hippocampus or cortex of wild-type mice (*Figure 6A*; *Figure 6—figure supplement 1*). Because this time-course matches the timing of synaptic GluN3A downregulation in vivo (*Henson et al., 2012*), we asked whether GluN3A expression influences GIT1/mTORC1 assembly. Biochemical analysis of GIT1 immunoprecipitates from hippocampi of P10 wild-type and *Grin3a*−/− showed that GluN3A removal enables the formation of GIT1/mTORC1 complexes at earlier stages, as judged by enhanced GIT1/mTOR and GIT1/Raptor binding (*Figure 6B*). Reciprocally, reexpression of full-length GluN3A (but not the GluN3A1082Δ mutant) in *Grin3a*−/− neurons was sufficient to prevent the GIT1/mTOR association, indicating that GluN3A-bound GIT1 cannot incorporate into the complex (*Figure 6C*). Taken together, the results support a model where GluN3A expression regulates the abundance of synaptic GIT1/mTORC1 complexes by directly binding GIT1, impeding its association with mTOR and limiting mTORC1 activation and downstream protein synthesis of plasticity factors. Conversely, developmental or genetic GluN3A downregulation enables GIT1/mTORC1 formation and primes synapses for mTORC1-dependent translation (*Figure 6D*).

To test this model, we asked whether genetic manipulations of GluN3A/GIT1 interactions affect the timing and magnitude of mTORC1-dependent protein synthesis. A first set of experiments showed that protein synthesis in young cortical neurons (DIV7–9) is not dependent on mTORC1 activation,

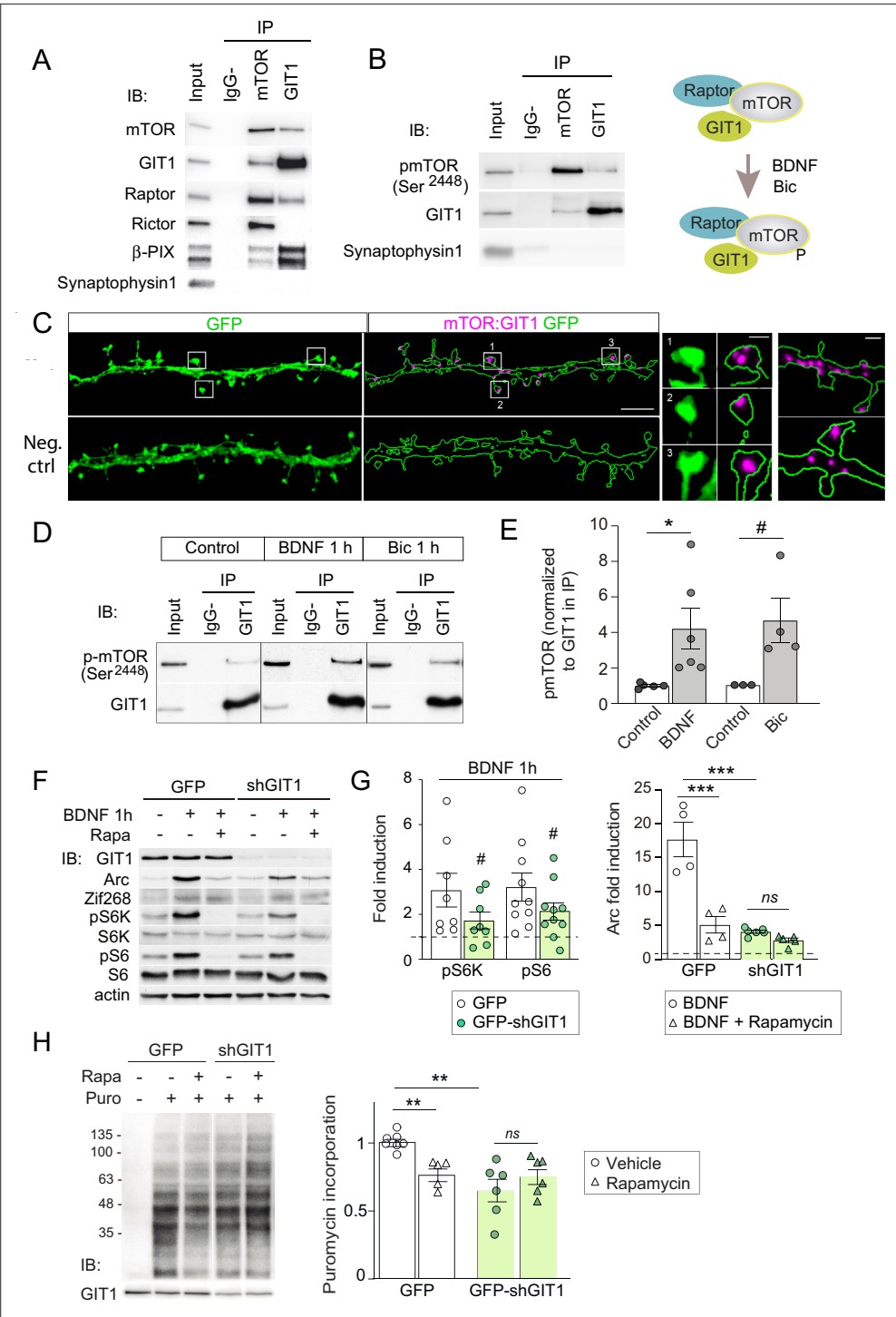

**Figure 5.** GIT1/mechanistic target of rapamycin (mTOR)/Raptor complexes couple synaptic activation to mTORC1-dependent protein synthesis. (**A, B**) Protein extracts from P16 mouse hippocampus were solubilized with 0.3 % CHAPS buffer, incubated with antibodies against mTOR or GIT1 (IP), and immunoprecipitated proteins analysed by immunoblot (IB). Input: 10 % of lysate used for immunoprecipitation. IgG–: no antibody control. A cartoon of the interactions and regulation by activity (see panel **D**) is shown. (**C**) Representative images of proximity ligation assay for rat mTOR: GIT1 (magenta) in days in vitro (DIV) 17 rat hippocampal neurons transfected with GFP (green) to visualize dendritic morphology (scale bar, 5 µm). High magnification examples of spines and dendrites (scale bars, 0.5 and 1 µm) are shown. As negative control, only mTOR primary antibody was used. (**D, E**) Rat cortical neurons stimulated with BDNF or bicuculline were solubilized with 0.3 % CHAPS and incubated with GIT1 antibody

*Figure 5 continued on next page*

*Figure 5 continued*

(IP). Representative immunoblots (**D**) and quantification of mTOR phosphorylation in GIT1 immunoprecipitates (**E**) are shown (*n* = 3–6 from three independent cultures; *p < 0.05, # = 0.06, two-tailed unpaired *t*-test). (**F–H**) Primary mouse cortical neurons were infected with lentiviruses expressing GFP or GFP-shGIT1 on DIV7. mTOR responses to BDNF (**F, G**) and puromycin incorporation (**H**) in the presence or absence of 100 nM rapamycin were analyzed at DIV14. Quantification of phosphorylated S6K and S6 and Arc induction (#pS6K: p = 0.13, #pS6: p = 0.05, two-tailed paired; ***p < 0.001, two-way analysis of variance [ANOVA] followed by Tukey's test) (**G**) and puromycin levels normalized to Ponceau S (*n* = 5–7 from four independent cultures, **p < 0.01, two-way ANOVA followed by Tukey's test) (**H**) are shown.

The online version of this article includes the following figure supplement(s) for figure 5:

**Source data 1.** Coimmunoprecipitation assays of GIT1 with mechanistic target of rapamycin (mTOR), Raptor, and Rictor in P16 mouse hippocampus.

**Source data 2.** Coimmunoprecipitation of GIT1 with phosphorylated mechanistic target of rapamycin (mTOR) in Ser$^{2448}$ in P16 mouse hippocampus.

**Source data 3.** Coimmunoprecipitaion of GIT1 and phosphorylated mechanistic target of rapamycin (mTOR) in days in vitro (DIV) 17 hippocampal neurons after bicuculline and BDNF treatment.

**Source data 4.** Western blots of mechanistic target of rapamycin (mTOR) effectors and immediate-early gene (IEG) induction by BDNF in the presence or absence of rapamycin in days in vitro (DIV) 14 cortical neurons infected with control or shGIT1-expressing lentiviruses.

**Source data 5.** Western blots of puromycin incorporation in days in vitro (DIV) 14 cortical neurons infected with control or shGIT1-expressing lentiviruses.

with strong rapamycin sensitivity emerging at later stages (DIV14) (*Figure 6—figure supplement 2*). Knockdown of GluN3A resulted in a large increase in protein synthesis in DIV7–9 neurons, which exhibited a rapamycin dependence typical of mature neurons (*Figure 6E*). Robust rapamycin-dependent protein synthesis was also observed in *Grin3a−/−* neurons (*Figure 6F*). Reexpression of GluN3A, but not GluN3A1082Δ, reduced protein synthesis rates and was sufficient to block mTORC1 dependence, reinstating a juvenile mode of protein synthesis (*Figure 6F*). Thus GluN3A, via binding to GIT1, controls the age-dependent switch between mTORC1-independent and mTORC1-dependent protein synthesis.

## Long-term memory formation is enhanced in *Grin3a*-deficient mice in a rapamycin-dependent manner

While GluN3A expression is typical of immature synapses at early postnatal stages as illustrated in our model, electron microscopy analyses demonstrate that subsets of synapses continue to express GluN3A into adulthood in areas such as the hippocampal CA1 (*Roberts et al., 2009*); and a recent mRNA expression study revealed that significant GluN3A levels are retained in a variety of brain regions (*Murillo et al., 2021*). Previous work showed that transgenic GluN3A overexpression impairs memory consolidation in hippocampal-dependent paradigms such as the Morris water maze (*Roberts et al., 2009*), but whether endogenous GluN3A expression has a physiological role in memory formation is unknown. We hypothesized that GluN3A modulation of synaptic mTORC1 signaling might provide a mechanism to set modes of translational control participating in memory encoding.

We reasoned that, if so, GluN3A deletion would create a permissive environment for stable memory formation and tested this by assessing mice learning in increasingly demanding tasks. Testing of *Grin3a−/−* mice in a standard version of the Morris water maze (four trials per day) did not reveal differences in the latencies to reach the hidden platform relative to wild-type controls (*Figure 7— figure supplement 1*). Wild-type and *Grin3a−/−* mice displayed similar preferences for the target quadrant in probe trials where the platform was removed from the pool at the end of training, confirming that both had learnt the platform location (PT1; *Figure 7—figure supplement 1C*). Differences emerged with a more demanding version of the task (two trials per day): both male and female *Grin3a−/−* mice reached the platform significantly faster than wild types, with shorter latencies by day 3 of training and greater preference for the target quadrant in probe trials (PT1; *Figure 7A and B*; *Figure 7—figure supplement 1B*, D). No differences were observed in a visible version of the maze or in swim velocities, suggesting that motor or perceptual differences do not account for the phenotype (*Figure 7—figure supplement 1E*, F).

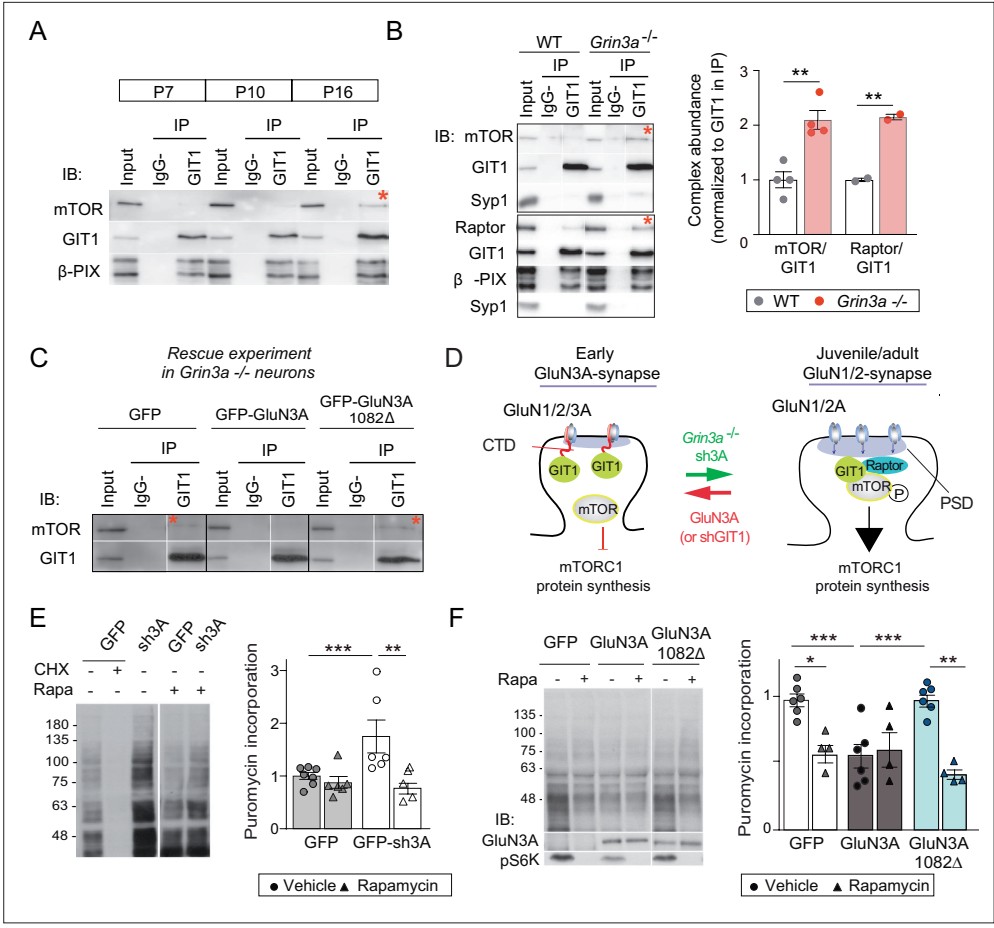

**Figure 6.** GluN3A/GIT1 interactions control the age-dependent onset of mTORC1-dependent protein synthesis.
(**A**) Hippocampi from P7, P10, and P16 wild-type mice were lysed, immunoprecipitated with GIT1 antibody and probed for the indicated antibodies. Input: 10 % of the lysate used for immunoprecipitation. IgG−: negative control without antibody. Red asterisks here and other panels indicate mechanistic target of rapamycin (mTOR)- and Raptor bound to GIT1. (**B**) GIT1/mTORC1 complex formation is enhanced in P10 *Grin3a−/−* hippocampus. Representative blots of GIT1 immunoprecipitates and quantifications are shown (*n* = 2–4 mice; **p < 0.01 unpaired *t*-test). Bound mTOR and Raptor are normalized to immunoprecipitated GIT1. Syp1: synaptophysin 1. (**C**) *Grin3a−/−* cortical neurons infected with GFP, GFP-GluN3A, or GFP-GluN3A1082Δ were solubilized and GIT1 immunoprecipitates blotted as indicated (IB). (**D**) GIT1/GluN3A control mTORC1 translation. Left: at early postnatal stages, immature synapses express GluN3A-NMDARs, which bind the postsynaptic scaffold GIT1 via their C-terminal tail preventing the nucleation of GIT1/mTORC1 and the mTORC1-mediated synthesis of plasticity proteins. Right: at juvenile/adult stages, GluN3A downregulation enables GIT1/mTOR/Raptor complex assembly and primes synapses for mTORC1 translation of mRNAs involved in synapse and memory consolidation. The genetic manipulations shown here to alter the age-dependent switch from mTORC1-independent to mTORC1-dependent modes of translation are indicated. Note that GluN3A expression is retained by subsets of synapses in adult brains and might play roles in selecting synapses that will be recruited to stably encode memory traces (see Discussion). (**E**) Mouse cortical neurons were infected with lentiviruses expressing GFP or GFP-sh3A on days in vitro (DIV) 3 and protein synthesis analyzed at DIV7–9. Representative blots and quantification of puromycin incorporation in infected neurons treated with rapamycin (100 nM), cycloheximide (CHX, 25 µM), or vehicle.
(**F**) *Grin3a−/−* cortical neurons were infected with GFP, GFP-GluN3A, or GFP-GluN3A1082Δ lentiviruses, and protein synthesis analyzed at DIV12. GluN3A expression and mTOR activation were monitored with the indicated antibodies (IB). In panels D–E, *n* = 4–7 from three to four independent cultures (*p < 0.05, **p < 0.01, ***p < 0.001, two-way analysis of variance [ANOVA] followed by Tukey's test).

The online version of this article includes the following figure supplement(s) for figure 6:

**Source data 1.** Coimmunoprecipitation of GIT1 and mechanistic target of rapamycin (mTOR) in lysates from P7, P10, and P16 mouse hippocampus.

*Figure 6 continued on next page*

*Figure 6 continued*

**Source data 2.** Coimmunoprecipitation of GIT1 with mechanistic target of rapamycin (mTOR) and Raptor in hippocampal lysates from P10 wild-type and *Grin3a−/−* mice.

**Source data 3.** Coimmunoprecipitation of GIT1 with mechanistic target of rapamycin (mTOR) in *Grin3a−/−* cortical neurons infected with GFP, GFP-GluN3A, and GFP-GluN3A1082Δ lentiviruses.

**Source data 4.** Western blots of puromycin incorporation in neurons infected with control or sh3A-expressing lentiviruses.

**Source data 5.** Western blots of puromycin incorporation in the presence or absence of rapamycin in *Grin3a−/−* cortical neurons infected with GFP, GFP-GluN3A, and GFP-GluN3A1082Δ.

**Figure supplement 1.** Postnatal regulation of GIT1/mTORC1 complexes in mouse somatosensory cortex.

**Figure supplement 1—source data 1.** Annotated western blots and original scans.

**Figure supplement 2.** Age-dependent emergence of mTORC1-dependent protein synthesis in cultured rat cortical neurons.

**Figure supplement 2—source data 1.** Annotated western blots and original scans.

---

Similarly reduced learning thresholds had been reported in mice with elevated activity of mTOR or other pathways controlling translation (*Banko et al., 2007*; *Costa-Mattioli et al., 2007*; *Hoeffer et al., 2008*; *Stern et al., 2013*), often at the expense of impaired ability to respond to changed environments, altered memory fidelity, or appearance of perseverant and repetitive behaviors (*Banko et al., 2007*; *Hoeffer et al., 2008*; *Santini et al., 2013*; *Shrestha et al., 2020b*; *Trinh et al., 2012*). Thus, we evaluated cognitive flexibility by retraining the mice to learn a new platform location ('reversal'). *Grin3a−/−* mice were better at shifting their preference relative to wild-type controls as evident in probe trials conducted 7 days after reversal (PT2; *Figure 7A and B*, *Figure 7—figure supplement 1B*, D). No perseverative behavior was observed either in a Y-maze spontaneous alternation task (*Figure 7—figure supplement 1G*). These results showed that GluN3A deletion facilitates spatial learning and memory without the unwanted effects associated to other modulators of translation.

We then assessed associative memory formation using two tasks that depend on new protein synthesis and can be achieved with the single pairing of a conditioned (CS) and unconditioned stimulus (US): conditioned taste aversion (CTA) and contextual fear conditioning (FC). In CTA, a novel taste (saccharin, CS) is associated with an aversive US (LiCl, which induces nausea). The LiCl dose (US) and temporal contiguity between CS–US can be regulated to evaluate standard memory (*Figure 7C*), or 'enhanced' memory by using a weaker paradigm (*Figure 7F*; *Adaikkan and Rosenblum, 2015*). Transgenic mice with prolonged GluN3A expression into adulthood (dt GluN3A) displayed deficits in a standard CTA paradigm (US, LiCl 0.15 M i.p.) as judged by their similar preference for saccharin 24 hr after saline or LiCl injections (*Figure 7D*, green bars). This result was inline with the memory deficits reported in other behavioral paradigms (*Roberts et al., 2009*). Control experiments ruled out the possibility that the defect was due to insensitivity to LiCl or to defects in distinguishing flavors (*Figure 7—figure supplement 2*). By contrast, *Grin3a−/−* mice did not show differences relative to wild types in a standard paradigm of CTA memory (*Figure 7E*). We then used a weak CTA paradigm where the strength of the US was reduced (LiCl 0.025 M), and US–CS were separated by 5 hr (*Figure 7F*). Under these conditions, only *Grin3a−/−* mice formed an association between CS–US, as shown by their significantly reduced preference for saccharin after LiCl injection but intact preference in wild-type controls. The negative association was long-lasting as it could be observed 24 (*Figure 7G*) and 48 hr after conditioning (data not shown). To determine whether it was mTOR dependent, we treated mice with a subthreshold dosing regime of rapamycin (*Stoica et al., 2011*) that does not affect standard CTA memory (*Figure 7—figure supplement 2*). Rapamycin erased the weak CTA memory in *Grin3a−/−* mice (*Figure 7H*), supporting the notion that disinhibited mTOR signaling causes the cognitive enhancement.

In contextual FC, a particular environment (CS) is associated with a foot-shock (US) (*Figure 8A*). Wild-type and *Grin3a−/−* littermates showed similar freezing responses before the delivery of the foot-shock (*Figure 8B*). However, freezing was significantly stronger in *Grin3a−/−* mice 24 and 48 hr after a weak training protocol (single pairing of a tone with a 0.3 mA foot-shock, *Figure 8B and C*), demonstrating enhanced and lasting memory formation. No differences were observed in short-term (1 hr) memory that is protein synthesis independent (*Figure 8B*). As in CTA, rapamycin occluded the

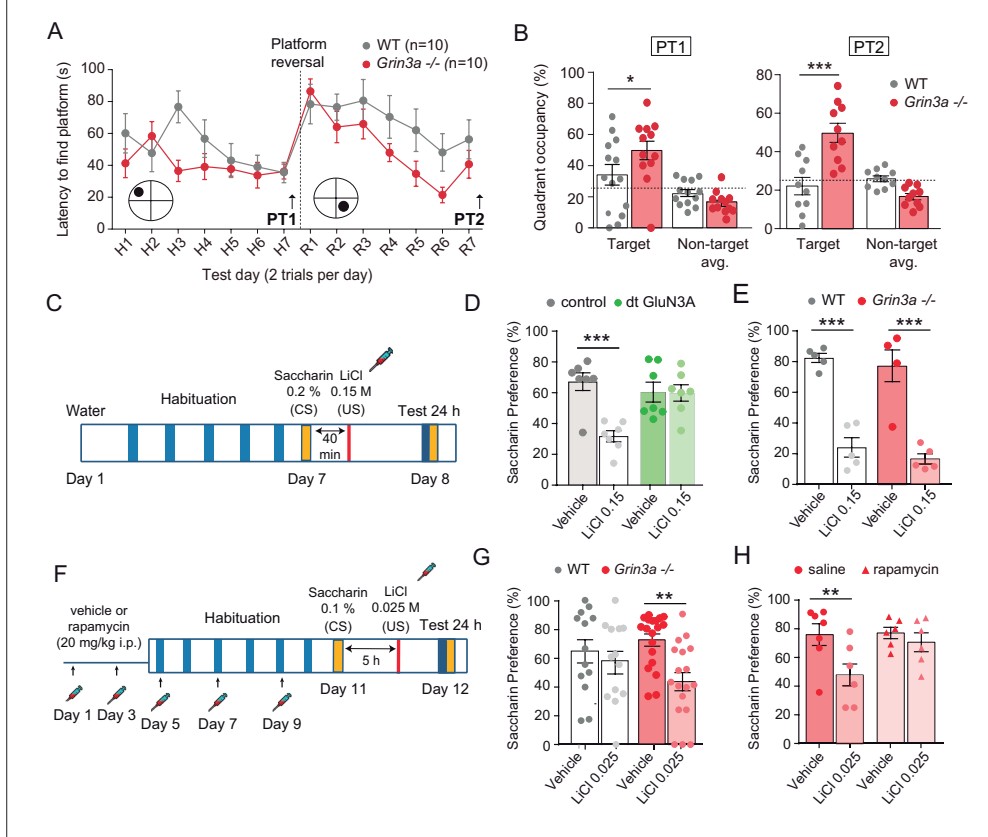

**Figure 7.** GluN3A deletion facilitates spatial and associative learning. (**A**) Escape latencies of male wild-type (WT) and *Grin3a*−/− mice on a weak version of the Morris water maze (two trials per day) during 7-day training and after platform reversal on day 8. (**B**) Probe trials performed 24 hr after day 7 (PT1), or 24 hr after reversal training (PT2) (*n* = 10–13 mice per genotype; two-way analysis of variance [ANOVA] with Bonferroni post hoc test, *p < 0.05, ***p < 0.0001). (**C**) Conditioned taste aversion (CTA) paradigm. (**D**) Saccharin preference of control and double transgenic (dt) GluN3A mice, and (**E**) WT and *Grin3a*−/− mice after vehicle or LiCl injection (*n* = 5–7 mice per group; ***p < 0.001, two-way ANOVA followed by Bonferroni post hoc test). (**F–H**) Weak CTA paradigm and rapamycin treatment regime. Decreased saccharin preference of *Grin3a*−/− mice on the weak CTA (**G**) was reversed by rapamycin (**H**) (**p < 0.01, two-way ANOVA followed by Bonferroni post hoc test).

The online version of this article includes the following figure supplement(s) for figure 7:

**Figure supplement 1.** Behavior of male and female *Grin3a*−/− mice in the Morris water maze.

**Figure supplement 2.** Controls for conditioned taste aversion (CTA) experiments.

difference between wild-type and *Grin3a*−/− mice (***Figure 8C***). Taken together, our behavioral results demonstrate that GluN3A deletion lowers the threshold for stable memory storage and provide pharmacological evidence linking the enhanced learning to a relief of GluN3A constraints on mTORC1 signaling.

Yet the cognitive enhancement could have been due to lack of GluN3A during development rather than adult stages. Also, GluN3A is expressed by excitatory neurons and somatostatin interneurons, both recently implicated in protein synthesis-dependent memory consolidation (***Sharma et al., 2020***; ***Shrestha et al., 2020a***). We therefore selectively ablated *Grin3a* from excitatory neurons or somatostatin-expressing interneurons by crossing floxed *Grin3a* mice (*Grin3a*[f/f]) with mice that express Cre recombinase under the control of the Ca[2+] calmodulin kinase IIα (*Camk2a*-Cre[ERT2]) or somatostatin (*Sst*-Cre) promoter. The first strategy allowed conditional deletion of GluN3A at adult stages by injecting tamoxifen (***Figure 8D***). Biochemical analysis of adult hippocampal lysates confirmed effective deletion of GluN3A, and revealed that ~70 % and ~ 20 % of GluN3A protein is expressed by excitatory and somatostatin interneurons, respectively (***Figure 8—figure supplement 1***). We then subjected the mice to the weak FC protocol. Adult deletion of GluN3A from excitatory neurons was

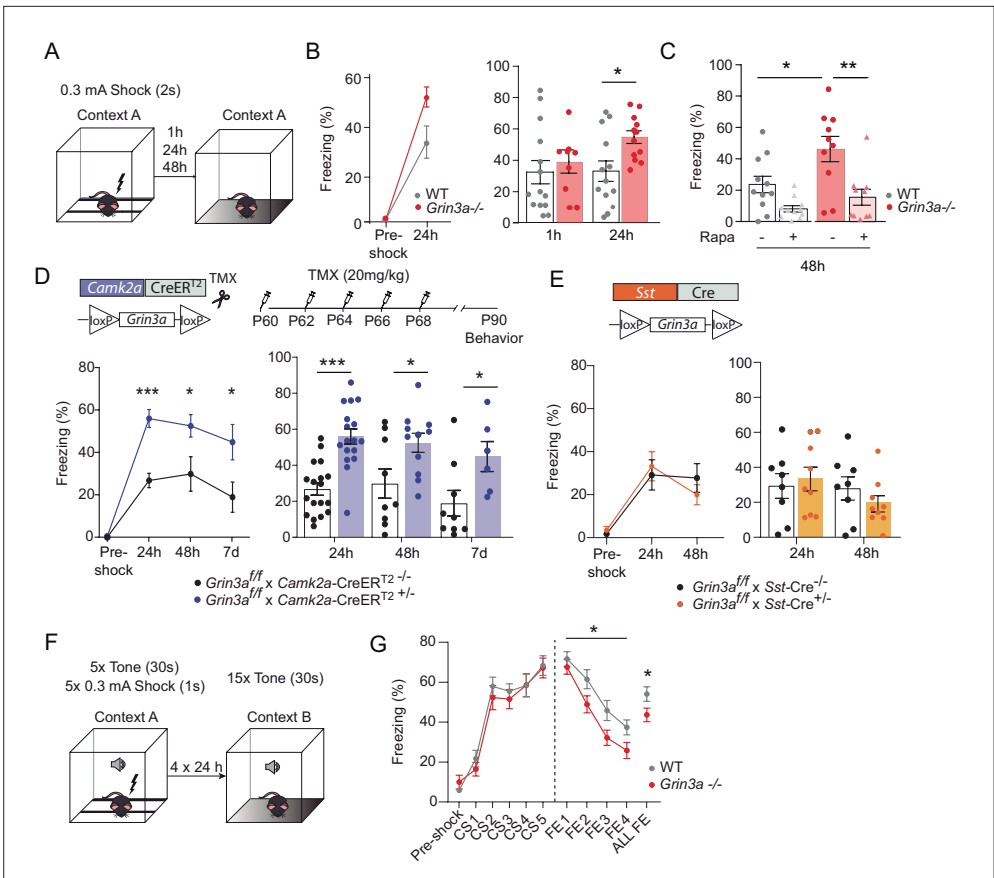

**Figure 8.** GluN3A deletion from excitatory neurons in adult mice is sufficient for memory enhancement. (**A**) Contextual fear conditioning test. (**B**) Enhanced contextual fear conditioning in *Grin3a−/−* mice 24 hr but not 1 hr after training (*n* = 9–13 mice per group; left: repeated measures two-way analysis of variance [ANOVA] with Bonferroni post hoc test, p = 0.004; right: two-tailed unpaired *t*-test, *p < 0.05). (**C**) Enhanced contextual fear conditioning in *Grin3a−/−* mice at 48 hr is reversed by rapamycin (2 × 20 mg/kg i.p., 24 hr apart, prior to training) (*n* = 10–11 mice per group; *p < 0.05; **p < 0.01, two-way ANOVA with Bonferroni post hoc test). (**D, E**) Conditional deletion of GluN3A from adult excitatory neurons, but not somatostatin (*Sst*) interneurons, enhances long-term contextual fear memory. The regime for tamoxifen (TMX) injection is indicated (*p < 0.05: ***p < 0.001 two-tailed unpaired *t*-test). (**F, G**) Cued-fear extinction in *Grin3a−/−* and wild-type littermates over a four-day fear extinction paradigm (*n* = 14–13 mice per group; *p = 0.0461 between-subjects effect, repeated measures ANOVA). Freezing levels were not different between phenotypes in FE1 (*t*(25) = 0.760, p = 0.455).

The online version of this article includes the following figure supplement(s) for figure 8:

**Figure supplement 1.** Expression of GluN3A and other synaptic proteins in conditional *Grin3a* knockout mice.

**Figure supplement 1—source data 1.** Annotated western blots and original scans.

**Figure supplement 1—source data 2.** Annotated western blots and original scans.

sufficient to enhance long-term memory, as shown by stronger freezing of *Grin3a*^f/f^ × *Camk2a*-Cre^ERT2^ mice 24, 48 hr, and even 7 days after training (***Figure 8D***). In contrast, *Grin3a*^f/f^ × *Sst*-Cre and control mice exhibited similar freezing levels 24 and 48 hr after training (***Figure 8E***). Thus, adult GluN3A expression in excitatory neurons gates long-term memory formation.

Finally, we evaluated fear extinction (FE), a form of memory where repeated presentation of a CS without reinforcement leads to the extinction of the acquired fear memory (***Andero and Ressler, 2012***). FE requires protein synthesis and is another indicator of behavioral flexibility that has been shown to be impaired after manipulation of general elements of translation. Mice were subjected to a strong auditory cued-FC protocol (five pairings of a tone [CS] with a 0.3 mA foot-shock) followed by four cued-FE sessions (15 CS alone, no foot-shock) (***Figure 8F***). Fear memory acquisition was similar between WT and *Grin3a−/−* littermates but FE was enhanced in *Grin3a−/−* mice (***Figure 8G***),

demonstrating that GluN3A deletion does not compromise the updating of memories but rather facilitates the extinction of fear memories.

## Discussion

In this study, we report a regulatory mechanism that affords spatiotemporal control of mTORC1-dependent translation in response to synaptic stimulation. Specifically we identify GIT1/mTORC1 complexes as key mediators of synaptic mTORC1 responses, and demonstrate that GluN3A-NMDARs, through direct association with GIT1, impede GIT1/mTORC1 assembly and negatively regulate synaptic mTORC1 activation and mTORC1-dependent translation. Using in vitro and in vivo genetic approaches, we further show that negative regulation by GluN3A determines the emergence of mature mTORC1-dependent protein synthesis in developing brains, and continues to play a role in adult life by placing boundaries on long-term memory storage. More broadly, our findings suggest that neuronal GIT1/mTORC1 complexes might provide a central site for the regulation and dysregulation of synaptic translation in other physiological and disease contexts.

### Modulation by GluN3A of GIT1/mTORC1 complex assembly

mTORC1 is a ubiquitous protein kinase complex that promotes protein synthesis and cell growth in response to a variety of signals including nutrient availability, energy levels, insulin, growth factors, and synaptic inputs. Coupling such diverse signals to mTORC1 activation requires regulated targeting to specific subcellular compartments. For instance, mTORC1 responses to amino acids require its recruitment by the Ragulator–Rag complex to lysosomal membranes, where interactions between positive (Rheb) and negative (Tsc1/2 complex) mTOR regulators take place (*Benjamin and Hall, 2014*; *Sancak et al., 2010*). Our observations suggest that GIT1 could play an analogous scaffolding role to position mTORC1 such that it senses synaptic signals, with negative regulation by GluN3A limiting mTORC1-dependent translation at specific developmental times and/or in specific subsets of synapses in adult brains.

First, GIT1/mTORC1 complexes are located at dendritic/synaptic sites and respond to synaptic stimuli, as shown by phosphorylation of GIT1-bound mTOR on Ser$^{2448}$, an event that is stimulated by NMDARs (*Figure 2—figure supplement 1*) and is amplified by feedback from the downstream mTORC1 substrate S6K (*Chiang and Abraham, 2005*). Second, knocking-down GIT1 blunts synaptic mTORC1 signaling and mTORC1-dependent translation of specific activity-regulated genes. Third, GIT1/mTORC1 abundance increases during postnatal development and is bidirectionally modulated by GluN3A expression. Fourth, the association of GIT1 with GluN3A is required for mTORC1 modulation, as demonstrated by the fact that expression in the *Grin3a* knockout of a GluN3A mutant lacking the GIT1-binding site does not rescue the increased assembly of GIT1 with mTOR (*Figure 6*) nor the increased activation of synaptic mTORC1 (*Figure 4*). Given that GluN3A and mTOR bind overlapping regions in GIT1 (*Fiuza et al., 2013*; *Smithson and Gutmann, 2016*), the most parsimonious explanation is that GluN3A competes with mTOR for binding to GIT1.

We previously reported that GluN3A modulates the formation of GIT1 complexes with βPIX (*Fiuza et al., 2013*), and might coordinately inhibit two central mechanism in spines that are necessary for memory consolidation —actin cytoskeletal rearrangements and protein synthesis. This action would be analogous to the translational repression by FMRP/CYFIP1 complexes (*De Rubeis et al., 2013*). Our results here (see *Figure 6B*) indicate that GluN3A exerts a more potent regulation over GIT1/mTORC1 than GIT1/βPIX complexes and suggest that mTOR modulation might be the primary event. Of note, rare coding variants in GIT1 have been identified in schizophrenic patients (*Kim et al., 2017*) and GIT1 knockout mice display deficits that resemble those seen in mice with elevated GluN3A expression, including reduced spine size and poor learning and memory (*Martyn et al., 2018*). Additional phenotypes reported in mice and flies upon GIT1 deletion, such as microcephaly, reduced neuronal size or hyperactivity, might also be related to mTOR modulation (*Hong and Mah, 2015*; *Won et al., 2011*).

Our RNAseq analyses indicate that GluN3A acts at the level of translation and would thus preserve the supply of activity-induced plasticity mRNAs but restrict their active translation to specific synapses, in contrast to classical NMDARs that work at both transcriptional and translational levels. Nevertheless, GluN3A knockdown in cultured neurons has been shown to enhance the transcription of a subset

of mRNAs (*Chen et al., 2020*) upon prolonged periods of synaptic activation (6–8 vs. 1 hr in the present study), and we cannot rule out later regulation by GluN3A of compensatory or homeostatic responses. Of note, tonic repression of mTOR-dependent protein synthesis by GluN2B-containing NMDARs has also been reported (*Wang et al., 2011*). However, the molecular determinants of stimulation or repression of protein synthesis were not addressed. It remains to be established whether GluN3A and GluN2B share common mechanisms.

## A role for GluN3A in restricting translation for precise circuit refinements and long-term memory storage

GluN3A-NMDARs are highly expressed during critical periods of experience-dependent neural circuit refinements, when they have been proposed to determine which synapses will be maintained or eliminated, and at lower levels in specific regions of the adult brain (*Murillo et al., 2021*). We propose a model whereby the lack or presence of GluN3A at postsynaptic sites contributes to spine-specific translation by setting an enhanced or repressed biochemical environment for mTORC1 signaling that will depend on the stage of brain development (*Figure 6D*) and the activity history of individual synapses. This is: synaptic GluN3A levels are downregulated by sensory experience and can be controlled at the level of individual synapses by activity-dependent endocytosis (*Pérez-Otaño et al., 2006*). The removal of GluN3A-NMDARs from active synapses would drive formation of nearby GIT1/mTORC1 complexes. This would locally increase the potential for dendritic translation of activity-regulated mRNAs, giving active inputs an advantage for consolidation versus less-active neighbors. Hence, competition for active mTORC1 would provide a means for selective synapse stabilization and memory storage. Defects in mTORC1 regulation might permit the consolidation of otherwise lost synaptic changes.

Such a competition-based model is supported by the localization of GluN3A to subsets of adult synapses (*Roberts et al., 2009*). It is also supported by the observations that in *Grin3a−/−* mice, the levels of GIT1/mTORC1 are increased and these mice exhibit enhanced capacity for memory storage, as shown by their performance in weak training protocols that are normally insufficient for stable memory formation in wild-type mice. Importantly, the restriction of dendritic translation to sites near active synapses is thought to underlie phenomena such as the competition between spines for lasting LTP expression (*Fonseca et al., 2004*) or the potentiation of synapses in clusters along the dendrite (*Fonseca et al., 2004*; *Govindarajan et al., 2011*). Incorporation into these models of the molecular components unveiled here might open avenues for testing how the above phenomena determine memory capacity and efficiency and for correcting cognitive dysfunction.

Our experiments using cell-type-specific and -inducible *Grin3a* knockout mice demonstrate a role of GluN3A in gating cognitive processing in the adult brain beyond its better recognized functions in postnatal neural circuit refinements, and identify excitatory neurons as the locus of GluN3A actions. In relation to memory, negative regulators are thought to provide an advantage by ensuring that only salient features are learnt and irrelevant events or associations are filtered out (*Abel et al., 1998*; *Cho et al., 2015*). For instance, temporal contiguity of events is required for many forms of associative learning; within the scale of seconds or minutes for classical conditioning paradigms, longer in other types of memory. In CTA, the CS and US can be hours apart, with temporal boundaries set by the strength of the US (*Adaikkan and Rosenblum, 2015*). Our results show that the absence of GluN3A broadens this temporal limit and facilitates learning of demanding tasks, i.e. where training is spaced apart or the presented stimuli are weaker. The reversal by rapamycin is consistent with the notion that the enhanced readiness of the mTORC1 translational machinery in GluN3A-deficient mice expands the range for consolidation of memory traces. While we used a subthreshold dose of rapamycin that does not alter memory or mTOR signaling in wild-type mice (*Stoica et al., 2011*), we cannot rule out potential nonsynaptic effects. Definitive proof will require the development of tools that directly disrupt GluN3A/GIT1 or GIT1/mTOR association or synaptic localization.

As far as tested here, GluN3A deletion does not impair other aspects of cognition such as memory flexibility or extinction. Yet significant GluN3A levels are retained in areas of the mouse and human adult brain with strong plasticity or functional integration needs (*Fulcher et al., 2019*; *Murillo et al., 2021*), and a recent study linked adult GluN3A expression to the control of emotional states (*Otsu et al., 2019*). In addition, genetic variations in *GRIN3A* have been shown to modulate prefrontal cortex activity (*Gallinat et al., 2007*) and episodic memory (*Papenberg et al., 2014*). Future investigations

should determine whether domains of cognition other than the ones we tested are compromised by GluN3A deletion.

## GluN3A and synaptic protein synthesis as selective therapeutic targets

The stabilization of memories requires de novo protein expression. Nevertheless, the effects on cognition of enhancing mTOR signaling or protein synthesis are perplexing. Loss of constraints on protein synthesis due to mutations in negative regulators of translation (*FMR1*, *MECP2*, or mTORC1 suppressors including *NF1*, *TSC1/2*, or the phosphatase *PTEN*) are associated with cognitive impairment and high incidence of autism spectrum disorders and intellectual disability (*Kelleher and Bear, 2008*), although a fraction of autistic individuals exhibit enhanced cognitive skills within specific domains (*Heaton and Wallace, 2004*). On the other hand, inhibiting the phosphorylation of eIF2α, which generally increases translation (*Costa-Mattioli et al., 2007*; *Stern et al., 2013*), or enhancing mTORC1 activity by removal of FKBP12 (*Hoeffer et al., 2008*) have been reported to lower memory thresholds. However, the cognitive enhancement came at the cost of reduced memory fidelity and cognitive flexibility even when cell-type-specific modulation was attempted (*Santini et al., 2013*; *Shrestha et al., 2020a*; *Trinh et al., 2012*), which we did not observe here. Key differences could be that other negative regulators of mTOR such as FMRP, PTEN, or Tsc1/2 are expressed in multiple cell types and neuronal locations, as demonstrated by their linkage to altered cell growth and appearance of tumors (*Lipton and Sahin, 2014*). Also, in some of the above situations, translation is constitutively activated and responses to incoming signals might be obliterated. By contrast, lack of GluN3A does not occlude mTORC1 activation but rather seems to prime mTOR activation by synaptic stimuli. At present, the enhancement of learning and memory produced by loss of GluN3A suggests that targeting GluN3A expression or signaling functions might be of therapeutic benefit. For instance, small molecules that perturb the GluN3A/GIT1 association might work in subtler ways than general translation regulators by specifically modulating synaptic mTORC1 signaling.

# Materials and methods
## Animals

Adult (3–6 months old) *Grin3a−/−*, *Grin3a*^tm1a(EUCOMM)Hmgu/H (*Grin3a^f/f*) and double-transgenic GFP-GluN3A (dtGluN3A) mice backcrossed for 10–12 generations into a C57Bl6/J background were used. Single transgenic mice were used as controls for dtGluN3A mice, and wild-type littermates from heterozygote crosses were controls for *Grin3a−/−* mice. Commercial C57BL6/J mice were purchased from Charles River Laboratories. For time-specific knockout of *Grin3a* in excitatory neurons, tamoxifen-inducible CaMKIIα-CreERT2^+/− mice (*Erdmann et al., 2007*) were crossed with *Grin3a^f/f* mice. Tamoxifen (Sigma-Aldrich T5648, 20 mg/ml dissolved in corn oil) was administered via oral gavage (five alternate days). For inhibitory neuron-specific knockout of *Grin3a*, Sst-IRES-Cre^+/− mice (JAX Stock No. 018973) were backcrossed with C57BL/6J mice for 12 generations and then bred with *Grin3a^f/f* mice. Male mice were used for behavioral experiments unless indicated. Animals were housed four to six per cage with ad libitum access to food and water and maintained in a temperature-controlled environment on a 12-hr dark/light cycle. All procedures were conducted in accordance with the European and Spanish regulations (2010/63/UE; RD 53/2013) and were approved by the Ethical Committee of the Generalitat Valenciana (2017/VSC/PEA/00196). For the cued-FC experiments, ethic protocols were approved by the Committee of Ethics of the Universitat Autònoma de Barcelona and the Generalitat de Catalunya.

## Primary neuronal cultures

Cortical and hippocampal neurons in primary culture were prepared as described (*Pérez-Otaño et al., 2006*). Briefly, cortices were dissected from E19 rat pups or E17.5 mice pups and dissociated with papain (Worthington Biochemical). Mouse primary neurons were used for rescue experiments in the *Grin3a*-null background shown in *Figures 4 and 6*, and for shGIT1 experiments in *Figure 5*. Neurons were plated at 75,000 cells per well on 12-well plates, 500,000 cells per well on 6-well plates and 1,000,000 cells/ dish on 60 mm dishes coated with laminin and poly-D-lysine and grown in Neurobasal Medium supplemented with B27 (Thermo Fisher).

Neurons were infected with lentiviruses 5 days prior to collection (timing of infection is indicated in figure legends). Neurotrophic factors and other drugs were used at the following concentrations: anisomycin (0.8 µM, Sigma-Aldrich A5892), recombinant human BDNF (100 ng/ml, PeproTech 450-02), bicuculline (50 µM, Abcam Ab120108), cycloheximide (25 µM, Sigma Aldrich C7698), (D,L)-APV (50 µM, Tocris 3693), MK801 (10 µM, Tocris 0924), rapamycin (100 nM, Alfa Aesar J62473), and puromycin (10 ng/ml, Sigma Aldrich P8833).

### Lentiviral vectors

For the generation of lentiviral constructs, full-length GluN3A and GluN3A1082Δ cDNAs were subcloned into a dual lentiviral vector Syn-WPRE-Syn-GFP kindly provided by Dr. Francisco G Scholl, University of Sevilla, Spain. For knockdown experiments, 19–20 base pairs (bp)-long small hairpin RNAs (shRNAs) directed to GluN3A (shGluN3A1185, target sequence: CTACAGCTGAGTTTAGAAA) or GIT1 (shGIT1, target sequence: TGATCACAAGAATGGGCATTA) were cloned into the pLentilox 3.7-GFP vector downstream the U6 promoter. The AAV encoding constitutively active human Rheb (AAV-caRheb, S16H) was kindly provided by Dr. Beverly Davidson, Children's Hospital of Philadelphia, University of Pennsylvania.

### RNA isolation, qRT-PCR, and RNAseq analyses

Total RNA from cultured cortical neurons was isolated using the Nucleospin RNA (Macherey-Nagel). RNA concentration and purity were assessed with NanoDrop. RNA quality was determined by the RNA Integrity Number (RIN) algorithm using the Agilent 2100 Bionalyzer Instrument; only samples with RIN >9 matched our standard.

For qRT-PCR experiments, first-strand cDNA was synthesized from 1 µg of total RNA using the Invitrogen SuperScript IV First-Strand cDNA Synthesis System (Thermo Fisher). Quantitative real-time PCR (qPCR) was performed using the Applied Biosystems QuantStudio 3 Real-Time PCR system and analyzed with the QuantStudio 3 Design and Analysis software (v1.5.1, Thermo Fisher). Briefly, real-time qPCR was assayed in a total volume of 20 µl reaction mixture containing the ready-to-use PyroTaq EvaGreen qPCR Mix Plus ROX (Cmb), 5 pmol of forward and reverse (rv) primers (detailed in key resource table) and cDNA. PCR thermal conditions included an initial hold stage with 5 min at 50 °C and 15 min at 95 °C followed by 40 cycles of denaturation for 30 s at 95 °C, annealing for 32 s at 60 °C and primer elongation for 32 s at 72 °C. All qPCR reactions were run in triplicates. Mean cycle threshold (Ct) values for each reaction were recorded and the relative RNA expression levels were calculated referring to *Gapdh*, encoding glyceraldehyde 3-phsophate dehydrogenase: $\Delta Ct = Ct\_GAPDH - Ct\_ (target\ gene)$. The gene expression fold change normalized to GAPDH and relative to control sample was calculated as $2^{\Delta Ct}$.

### Key resources table

| Reagent type (species) or resource | Designation | Source or reference | Identifiers | Additional information |
|---|---|---|---|---|
| Antibody | GIT-1 (mouse monoclonal, clone A-1) | Santa Cruz Biotechnology | Cat# sc-365084; RRID: AB_10850059 | PLA 1:150 |
| Antibody | Arc (mouse monoclonal, clone C-7) | Santa Cruz Biotechnology | Cat# sc-17839; RRID: AB_626696 | WB 1:100 |
| Antibody | beta-Tubulin III (mouse monoclonal) | Sigma-Aldrich | Cat# T8660; RRID: AB_477590 | WB 1:20,000 |
| Antibody | NMDAR1, all splice variants (mouse monoclonal, clone R1JHL) | Millipore | Cat# MAB1586; RRID: AB_11213180 | WB 1:1000 |
| Antibody | NR2B (mouse monoclonal, clone BWJHL) | Millipore | Cat# 05–920; RRID: AB_417391 | WB 1:1000 |
| Antibody | NR3A (mouse monoclonal) | Kindly provided by Jim Trimmer | N/A | WB 1:100 |
| Antibody | PSD-95 (mouse monoclonal, clone K28/43) | Antibodies Incorporated | Cat# 75–028 RRID: AB_10698024 | WB 1:1000 |
| Antibody | Puromycin (mouse monoclonal, clone 12D10) | Millipore | Cat# MABE343; RRID: AB_2566826 | WB 1:2000 |

*Continued on next page*

*Continued*

| Reagent type (species) or resource | Designation | Source or reference | Identifiers | Additional information |
|---|---|---|---|---|
| Antibody | Synapsin I (mouse monoclonal, clone 46.1) | Synaptic Systems | Cat# 106 011 RRID: AB_2619772 | WB 1:5000 |
| Antibody | Synaptophysin (mouse monoclonal, clone SY38) | Millipore | Cat# MAB5258-20UG; RRID: AB_11214133 | WB 1:2000 |
| Antibody | CREB (rabbit monoclonal, clone 48H2) | Cell Signaling Technology | Cat# 9197; RRID: AB_331277 | WB 1:1000 |
| Antibody | NR2A (rabbit monoclonal, clone A12W) | Millipore | Cat# 05–901 R; RRID: AB_10805961 | WB 1:1000 |
| Antibody | Phospho-CamKinase II alpha (CaMKII $\alpha$) Thr286 (rabbit monoclonal, clone D21E4) | Cell Signaling Technology | Cat# 12716; RRID: AB_2713889 | WB 1:1000 |
| Antibody | Phospho-p70 S6 kinase Thr389 (rabbit monoclonal, clone 108D2) | Cell Signaling Technology | Cat# 9234; RRID: AB_2269803 | WB 1:1000 |
| Antibody | Raptor (rabbit monoclonal, clone 24C12) | Cell Signaling Technology | Cat# 2280; RRID: AB_561245 | WB 1:1000 |
| Antibody | Rheb (rabbit monoclonal, clone E1G1R) | Cell Signaling Technology | Cat# 13879; RRID: AB_2721022 | WB 1:1000 |
| Antibody | Rictor (rabbit monoclonal, clone 53A2) | Cell Signaling Technology | Cat# 2114; RRID: AB_2179963 | WB 1:500 |
| Antibody | S6 ribosomal protein (rabbit monoclonal, clone 5G10) | Cell Signaling Technology | Cat# 2217; RRID: AB_331355 | WB 1:1000 |
| Antibody | GIT1 (rabbit polyclonal) | Cell Signaling Technology | Cat# 2919; RRID: AB_2109982 | IP 1:200, WB 1:1000 |
| Antibody | Egr-1/Zif268 (rabbit polyclonal) | Santa Cruz Biotechnology | Cat# sc-110; RRID: AB_2097174 | WB 1:500 |
| Antibody | beta-Pix, SH3 domain (rabbit polyclonal) | Millipore | Cat# 07–1450; RRID: AB_1586904 | WB 1:1000 |
| Antibody | c-Fos (rabbit polyclonal) | Santa Cruz Biotechnology | Cat# sc-52; RRID: AB_2106783 | WB 1:500 |
| Antibody | CaMKII $\alpha$ (rabbit polyclonal) | Sigma-Aldrich | Cat# C6974; RRID: AB_258984 | WB 1:1000 |
| Antibody | mTOR (rabbit polyclonal) | Cell Signaling Technology | Cat# 2972; RRID: AB_330978 | IP 1:100, PLA 1:150, WB 1:1000 |
| Antibody | NMDAR2A&B, pan antibody (rabbit polyclonal) | Millipore | Cat# AB1548; RRID: AB_11212156 | WB 1:1000 |
| Antibody | NR3A (rabbit polyclonal) | Millipore | Cat# 07–356; RRID: AB_2112620 | WB 1:1000 |
| Antibody | p30alpha (rabbit polyclonal) | Santa Cruz Biotechnology | Cat# sc-535; RRID: AB_632138 | WB 1:1000 |
| Antibody | p44/42 MAPK (Erk1/2) (rabbit polyclonal) | Cell Signaling Technology | Cat# 9102; RRID: AB_330744 | WB 1:1000 |
| Antibody | p70 S6 kinase (rabbit polyclonal) | Cell Signaling Technology | Cat# 9202; RRID: AB_331676 | WB 1:1000 |
| Antibody | Phospho-CREB Ser133 (rabbit polyclonal) | Millipore | Cat# 06–519; RRID: AB_310153 | WB 1:1000 |
| Antibody | Phospho-mTOR Ser2448 (rabbit polyclonal) | Cell Signaling Technology | Cat# 2971; RRID: AB_330970 | WB 1:1000 |
| Antibody | Phospho-p38 MAPK Thr180/Tyr182 (rabbit polyclonal) | Cell Signaling Technology | Cat# 9911; RRID: AB_10695905 | WB 1:1000 |
| Antibody | Phospho-p44/42 MAPK (Erk1/2) Thr202/Tyr204 (rabbit polyclonal) | Cell Signaling Technology | Cat# 9101; RRID: AB_331646 | WB 1:1000 |

*Continued on next page*

Continued

| Reagent type (species) or resource | Designation | Source or reference | Identifiers | Additional information |
|---|---|---|---|---|
| Antibody | Phospho-S6 ribosomal protein Ser240/244 (rabbit polyclonal) | Cell Signaling Technology | Cat# 2215; RRID: AB_331682 | WB 1:1000 |
| Cell line (*Homo sapiens*) | HEK293 | ATCC | Cat# CRL-1573; RRID: CVCL_0045 | |
| Chemical compound, drug | (−)-Bicuculline methiodide | Abcam | Cat# Ab120108; CAS: 55950-07-7 | |
| Chemical compound, drug | (D,L)-APV sodium salt | Tocris | Cat# 3693; CAS: 1303993-72-7 | |
| Chemical compound, drug | Anisomycin | Sigma-Aldrich | Cat# A5892; CAS: 22862-76-6 | |
| Chemical compound, drug | B27 supplement | Thermo Fisher Scientific | Cat# 17504044 | |
| Chemical compound, drug | BDNF | PeproTech | Cat# 450-02; AN: P23560 | |
| Chemical compound, drug | CGP-78608 | Tocris | Cat# 1493; CAS: 1135278-54-4 | |
| Chemical compound, drug | cOmplete Protease Inhibitor Cocktail | Sigma-Aldrich | Cat# 04693116001 | |
| Chemical compound, drug | Cycloheximide | Sigma-Aldrich | Cat# C7698; CAS: 66-81-9 | |
| Chemical compound, drug | MK-801 | Tocris | Cat# 0924; CAS: 77086-22-7 | |
| Chemical compound, drug | Puromycin dihydrochloride | Sigma-Aldrich | Cat# P8833; CAS: 58-58-2 | |
| Chemical compound, drug | Rapamycin | Alfa Aesar | Cat# J62473; CAS: 53123-88-9 | |
| Chemical compound, drug | Tamoxifen | Sigma-Aldrich | Cat# T5648 | |
| Chemical compound, drug | Tetrodotoxin citrate | Alomone Labs | Cat# T-550; CAS: 18660-81-6 | |
| Commercial assay, kit | Duolink In Situ Red Starter Kit Mouse/Rabbit | Sigma-Aldrich | Cat# DUO92101 | |
| Commercial assay, kit | MasterMix qPCR ROx PyroTaq EvaGreen | cmb | Cat# 87H24 | |
| Commercial assay, kit | Nucleospin RNA | Macherey-Nagel | Cat# 740955.50 | |
| Commercial assay, kit | Pierce BCA Protein Assay kit | Thermo Fisher Scientific | Cat# 23,227 | |
| Commercial assay, kit | SuperScript IV First-Strand cDNA Synthesis System | Invitrogen | Cat# 18-091-050 | |
| Genetic reagent (*Mus musculus*) | Mouse: B6;129 × 1-Grin3a$^{tm1Nnk/J}$ | The Jackson Laboratory | Cat# JAX:029974; RRID: IMSR_JAX:029974 | |
| Genetic reagent (*Mus musculus*) | Mouse: CaMKII $\alpha$ -CreERT2$^{+/-}$ | *Erdmann et al., 2007* | | |
| Genetic reagent (*Mus musculus*) | Mouse: Grin3a$^{tm1a(EUCOMM)Hmgu}$/H | EUCOMM | | |
| Genetic reagent (*Mus musculus*) | Mouse: Sst-IRES-Cre | The Jackson Laboratory | Stock: 018973 | |
| Genetic reagent (virus) | LV-hSYN-WPRE-hSYN-GFP-WPRE | *Gascón et al., 2008* | | |
| Genetic reagent (virus) | LV-hSYN-GluN3A-WPRE-hSYN-GFP-WPRE | This paper | | See Materials and methods; generated/stored in Perez-Otano's lab. |
| Genetic reagent (virus) | LV-hSYN-GluN3A1082Δ-WPRE-hSYN-GFP-WPRE | This paper | | See Materials and methods; generated/stored in Perez-Otano's lab. |

*Continued on next page*

*Continued*

| Reagent type (species) or resource | Designation | Source or reference | Identifiers | Additional information |
|---|---|---|---|---|
| Genetic reagent (virus) | pLentiLox3.7-GFP (pLL3.7-GFP) | Kindly provided by Dr. Michael Ehlers | Addgene plasmid #11795; RRID: Addgene_11795 | |
| Genetic reagent (virus) | pLL3.7-shGluN3A1185-GFP (Target sequence: CTACAGCTGAGTTTAGAAA) | *Yuan et al., 2013* | | |
| Genetic reagent (virus) | pLL3.7-shGIT1-GFP (Target sequence: TGATCACAAGAATGGGCATTA) | This paper | | See Materials and methods; generated/ stored in Perez-Otano's lab. |
| Recombinant DNA reagent (plasmid) | pcDNA1-Amp-GluN1-1A | *Perez-Otano et al., 2001* | | |
| Recombinant DNA reagent (plasmid) | pcDNA1-Amp-GluN2A | *Perez-Otano et al., 2001* | | |
| Recombinant DNA reagent (plasmid) | pCIneo-GFPGluN3A | *Perez-Otano et al., 2001* | | |
| Recombinant DNA reagent (plasmid) | pCIneo-GFPGluN3A1082Δ | This paper | | See Materials and methods; generated/ stored in Perez-Otano's lab. |
| Recombinant DNA reagent (plasmid) | pRK5-GFP | Kindly provided by Dr. Michael Ehlers | | |
| Sequence-based reagent (oligonucleotide) | *Arc*_fwd (mouse) | This paper | | GAGCCTACAGAGCCAGGAGA |
| Sequence-based reagent (oligonucleotide) | *Arc*_rv (mouse) | This paper | | TGCCTTGAAAGTGTCTTGGA |
| Sequence-based reagent (oligonucleotide) | *c-Fos*_fwd (mouse/rat) | *Chen et al., 2020* | | CTGCTCTACTTTGCCCCTTCT |
| Sequence-based reagent (oligonucleotide) | *c-Fos*_rv (mouse/rat) | *Chen et al., 2020*; | | TTTATCCCCACGGTGACAGC |
| Sequence-based reagent (oligonucleotide) | *GAPDH*_fwd (mouse/rat) | This paper | | CATGGCCTTCCGTGTTCCT |
| Sequence-based reagent (oligonucleotide) | *GAPDH*_rv (mouse/ rat) | This paper | | TGATGTCATCATACTTGGCAGGTT |
| Software, algorithm | ImageJ | Schneider, Rasband and Eliceiri, 2012 | https://imagej.nih.gov/ij/ | |
| Software, algorithm | ImageQuant software version 5.2 | GE Healthcare | | |
| Software, algorithm | Prism software version 7.00 | Graphpad | | |
| Software, algorithm | QuantStudio 3 Design and Analysis software v1.5.1 | Thermo Fisher Scientific | | |
| Software, algorithm | SMART software for video-tracking | PanLab S.L. | | |

For RNAseq experiments, we performed bulk mRNA sequencing single end with a length of 50 bp using the RNAseq Illumina Hiseq2500. The preparation of the polyA sequencing library, library's quality control and quantification, sequencing run and base calling data were carried out by the Genomics Core Facility of the Centre for Genomic Regulation (CRG, Barcelona). For the analysis, adapters were trimmed using trim_galore v0.6.4_dev and reads with longer length than 40 bp were selected. Trimmed reads were aligned using star c2.6.1b to the mouse genome (mm10). Reads with mapq >30 were selected using Samtools v1.9. Mapped reads were quantified using R scripts (R version 4.0.3, 2020), Rsubread v2.4.3 and the Mus_musculus.GRCm38.99.gtf annotation data. Differential expression analysis was performed using DESeq2 1.31.1 and limma 3.46.0; genes were annotated using biomaRt v2.46.3 and Volcano plots were performed with EnhancedVolcano 1.6.0. The tracks from the samples were performed with DeepTools v3.5.0, normalized with RPKM and visualization was done in IGV v2.6.3.

## Protein extraction and western blotting

Cultured neurons were collected in lysis buffer containing 50 mM Tris–HCl pH 6.8, 10 % glycerol, 2 % SDS, 0.1 M (D,L)-dithiothreitol, 0.04 % bromophenol blue, and supplemented with protease

(cOmplete Protease Inhibitor Cocktail) and phosphatase (PhosSTOP) inhibitors. Lysates were incubated for 10 min at 65 °C, briefly centrifuged at maximum speed and proteins separated by SDS–PAGE. Proteins were transferred onto PVDF membranes (GE Healthcare). After incubation with primary antibodies, membranes were incubated with secondary HRP-conjugated secondary antibodies (1:10,000, GE Healthcare). Signals were visualized with film autoradiography or the Amerham 680 Blot Imager, and nonsaturated immunoreactive bands were quantified using the ImageQuant 5.2 software.

For in vivo studies on mouse tissue, hippocampi and somatosensory cortex were dissected on ice, snapped frozen in liquid nitrogen and stored at −80 °C until processing. Tissues were homogenized in 15 (wt/vol) volumes of modified ice-cold RIPA buffer (50 mM Tris–HCl pH 7.5, 150 mM NaCl, 1% NP-40, 0.05 % deoxycholate, 0.01 % SDS) supplemented with protease and phosphatase inhibitors, sonicated and centrifuged for 20 min at 16,200 × $g$ at 4 °C. Protein content was estimated using a Pierce BCA Assay kit (Thermo Fisher) before immunoblotting.

## Immunoprecipitation

Cultured cortical neurons or mouse hippocampus or somatosensory cortex were solubilized for 30 min in cold lysis buffer containing 0.1 % Triton X-100, 0.1 % SDS, 150 mM NaCl, 10 mM EDTA and 50 mM HEPES or 0.3 % CHAPS 3-[(3-cholamidopropyl)dimethylammonio]-1-propanesulfonic acid, 150 mM NaCl, 1 mM EDTA, and 40 mM HEPES, supplemented with protease and phosphatase inhibitors. Insoluble material was removed by centrifugation at 16,200 × $g$ for 15 min and 100–150 μg of the resulting supernatants were incubated overnight at 4 °C with or without (IgG−) the immunoprecipitating antibody. Lysates were then incubated with protein A/G magnetic beads (BioRad) for 2 hr at 4 °C. Beads were precipitated using a magnetic rack, washed thrice in lysis buffer and immunoprecipitated proteins were eluted with SDS sample buffer and analyzed by western blotting.

## Proximity ligation assay

Cultured neurons transfected with pRK5-GFP were fixed at DIV17 with 4 % PFA, 4 % sucrose in phosphate buffered Saline (PBS) (RT, 10 min), incubated with blocking solution and permeabilized. Cells were then incubated with rabbit polyclonal anti-mTOR antibody and mouse monoclonal anti-GIT1 antibody overnight at 4 °C, washed with PBS, and incubated for 1 hr with PLA secondary probes (anti-mouse Plus and anti-rabbit Minus, Olink Bioscience) at 37 °C. Cells were washed twice with Duolink II Wash Buffer A (Olink Bioscience) and incubated with the ligase (1:40; Olink Bioscience) in ligase buffer for 30 min at 37 °C. After additional washes with Buffer A, cells were incubated with DNA polymerase (1:80; Olink Bioscience) in amplification buffer for 100 min at 37 °C in the dark. Cells were then washed with Duolink II Wash Buffer B (Olink Bioscience) and incubated with chicken polyclonal anti-GFP for 1 hr at room temperature. After washing with PBS, cells were incubated with secondary goat anti-chicken-Alexa Fluor 488 for 1 hr at room temperature. Finally, cells were washed in PBS and mounted on slides with Fluoroshield mounting medium (Sigma-Aldrich). Fluorescence images were acquired by using Nikon A1 Ti2 system with a sequential acquisition setting at 1024 × 1024 pixels resolution; cells were randomly selected from different coverslips.

## Protein synthesis assays

Basal protein synthesis was measured using a SUnSET (surface sensing of translation) assay. Briefly, primary cortical cultures were treated with 10 ng/ ml of puromycin for 30 min and lysed as described above. Untreated neurons and neurons preincubated with the protein synthesis inhibitor cycloheximide (15 min before puromycin) were used as controls. Proteins were resolved by SDS–PAGE and analyzed by western blotting using an anti-puromycin antibody. Ponceau S staining was used as protein loading control.

## Behavioral analysis
### Morris water maze

Mice were trained to find a submerged platform in a circular tank (190 cm diameter) filled with opaque white water in two or four trials per day with 45 min intertrial intervals (ITIs). If mice did not find the platform in 120 s, they were kindly guided to it. The hidden platform was relocated to the opposite quadrant after 7 days of training for the reversal training phase. Sixty-second-long probe tests in which platform was removed were performed at the end of each phase (PT1, after initial hidden

platform learning; PT2, after reversal learning), and time spent in the target quadrant was compared to the average time spent in all other quadrants. Mice were tracked throughout the whole protocol using the video-tracking software SMART (Panlab S.L.).

## Y-maze spontaneous alternation

Mice were introduced in a three-armed Y-shaped maze and recorded for 5 min. Correct triad scores were noted when all three arms were sequentially entered. Alternation indices were calculated as correct triads/possible triads. Maze was cleaned between animals with a water-based soap solution.

## Conditioned taste aversion

Test was adapted from *Adaikkan and Rosenblum, 2015*. In brief, mice were trained to drink from two bottles of water for 6 days. On conditioning day, water was changed for 0.2 % (regular CTA) or 0.1 % (weak CTA) saccharin for 40 min (regular) or 5 hr (weak) after the exposure, mice were injected LiCl intraperitoneally at 0.15 M (regular) or 0.025 M (weak). Saccharin preference was evaluated 24 hr after injection. For unconditioned taste preference, mice were presented two drinking bottles for 48 hr: one contained water and the other one of the following solutions: sucrose 5 % (sweet), NaCl 75 mM (salty), quinine 300 µM (bitter), and HCl 0.03 M (sour). Bottle sides were switched after 24 hr to avoid potential side bias. Solution preference was evaluated at 48 hr. For assessing sensitivity to LiCl toxicity, 'lying on belly' behavior was registered after injection of LiCl (0.15 M) or saline. This behavior consists in a totally general suppression of activity, and location of the mouse in the corner of a cage. The activity was measured for 20 min.

## Fear conditioning and extinction

FC and FE procedures were carried out with a computerized Fear and Startle system (Panlab-Harvard, Barcelona, Spain). Tones and shocks were delivered and controlled using Freezing v1.3.04 software (Panlab-Harvard, Barcelona, Spain). The fear chambers consisted of a black methacrylate box with a transparent front door (25 × 25 × 25 cm) inside a sound-attenuating cubicle (67 × 53 × 55 cm). Animals were habituated to the chambers for 5 min/day during two consecutive days prior to FC. The chambers were carefully cleaned before and after each mouse.

For contextual FC, mice were placed in the fear chambers and allowed to explore a context (CS) (metal grid floor, no light source) for 2 min. Mice were then presented with a tone (30 s, 2.8 kHz, 85 dB tone) that coterminated with a foot-shock (US) (0.3 mA, 2 s). Sixty seconds later, they were returned to their home cage. Conditioning was assessed at 1 (short-term memory), 24, and 48 hr or 7 days (long-term memory) by reintroducing mice in the conditioning context for 5 min. Freezing behavior, a rodent's natural response to fear defined as the absence of movement except respiration, was scored by a high sensitivity weight transducer system located at the bottom of the experimental chambers which records and analyses the signal generated by the movement of the animal.

For cued FC, mice were placed in the fear chambers for 5 min and then received five trials of a tone (CS) (30 s, 6 kHz, 75 dB) that coterminated with a foot-shock (US) (0.3 mA, 1 s). The ITI was of 3 min, and three additional minutes followed the last trial. The FE sessions were performed four times in consecutive days (FE1, FE2, FE3, and FE4) starting 24 hr after FC. For FE, mice were placed in the fear chambers for 5 min and then exposed to 15 trials of the 30 s CS tone alone (cued-fear) with a 30 s of ITI interval. An additional 30-s interval followed the last trial of FE. Different contexts were used for FC and FE tests. FC context consisted of a yellow light source (~10 lux), a grid floor of 25 bars (3 mm Ø and 10 mm between bars), a background noise of 60 dB produced by a ventilation fan and soapy water in a solution of ethanol 70 % was used for cleaning between sessions. FE context consisted of a red-light source (~10 lux), a grey plexiglass floor covering the bars, no background noise and soapy water in a solution of isopropyl alcohol 40 % was used as cleaning agent between sessions. Different routes were used to transport animals from their home cages to testing room in FC and FE days. Freezing levels were scored and averaged in 30-s slots.

## Electrophysiology

HEK293 cells were cultured, transfected, and recorded as previously described using GluN1A and GluN2A in pcDNA1/Amp and GFP-tagged GluN3A or GluN3A1082Δ subcloned in pCI-neo (*Chowdhury et al., 2013*). HEK293 cells were obtained from ATCC, and no mycoplasm contamination was

detected by regular testing. Briefly, cells were transfected with GluN1-1A, GluN2A, and either GFP-GluN3A or GFP-GluN3A1082Δ in a 1:1:3 ratio and maintained in medium with APV (250 μM). GFP was used as a transfection marker in cells where GluN3A constructs were omitted. Whole-cell recordings were made with on GFP-positive cells using a Multiclamp 700 A amplifier (Molecular Devices) 24 hr following transfection. Patch pipettes (2–4 MΩ) contained (in mM): 140 Cs methanesulfonate, 10 HEPES, 5 adenosine triphosphate (Na salt), 5 $MgCl_2$, 0.2 $CaCl_2$, and 10 BAPTA (pH 7.4). The extracellular solution contained (in mM) 150 NaCl, 5 KCl, 2 or 10 $CaCl_2$, 10 HEPES, 10 glucose (pH 7.4), and was adjusted to 330 mOsm with sucrose. Currents were digitized at 2 kHz and filtered at 1 kHz. Series resistance (90–95%) and whole-cell capacitance compensation were employed. Experiments were performed at a holding potential of –80 mV with ramps (300 ms to +50 mV) elicited following a 3 s application of glutamate (1 mM) and glycine (100 μM) at 20 °C. The $\Delta E_{rev}$, was calculated by subtracting the $E_{rev}$ obtained in 2 mM $Ca^{2+}$ from the $E_{rev}$ measured in 10 mM $Ca^{2+}$ and corrected for the junction potential between solutions. Initial peak currents were obtained from 1 s agonist applications in 2 mM $Ca^{2+}$ and used to calculate the current density. Experiments on glycine-gated diheteromeric GluN1/GluN3A receptors expressed in HEK293 cells were performed as previously described (*Grand et al., 2018*) using GluN1-1a and GFP-GluN3A or GFP-GluN3A1082Δ subcloned in pCI-neo (see above).

## Statistical analysis

Statistical analyses were conducted with GraphPad Prism software. Comparison of quantitative variables between two groups was performed using Student's *t*-test. One- or two-way analysis of variance (ANOVA) followed by a post hoc comparison test were used when more than two groups were compared, as indicated in the corresponding figure legend. Results are presented as mean ± standard error of the mean (SEM). Statistical methods used for behavioral studies are indicated in the corresponding figure legends.

## Acknowledgements

We thank Stuart Lipton and Nobuki Nakanishi for providing the *Grin3a* knockout mice, Beverly Davidson for the AAV-caRheb, Jose Esteban for help with behavioral and biochemical experiments, and Noelia Campillo, Rebeca Martínez-Turrillas, and Ana Navarro for expert technical help. Work was funded by the UTE project CIMA; fellowships from the Fundación Tatiana Pérez de Guzmán el Bueno, FEBS, and IBRO (to M.J.C.D.), Generalitat Valenciana (to O.E.-Z.), Juan de la Cierva (to L.G.R.), FPI-MINECO (to E.R.V., to S.N.) and Intertalentum postdoctoral program (to V.B.); ANR (GluBrain3A) and ERC Advanced Grants (#693021) (to P.P.); Ramón y Cajal program RYC2014-15784, RETOS-MINECO SAF2016-76565-R, ERANET-Neuron JTC 2019 ISCIII AC19/00077 FEDER funds (to R.A.); RETOS-MINECO SAF2017-87928-R (to A.B.); an NIH grant (NS76637) and UTHSC College of Medicine funds (to S.J.T.); and NARSAD Independent Investigator Award and grants from the MINECO (CSD2008-00005, SAF2013-48983R, SAF2016-80895-R), Generalitat Valenciana (PROMETEO 2019/020)(to I.P.O.) and Severo-Ochoa Excellence Awards (SEV-2013-0317, SEV-2017-0723).

## Additional information

### Funding

| Funder | Grant reference number | Author |
| --- | --- | --- |
| H2020 European Research Council | ERC 693021 | Pierre Paoletti |
| Ministerio de Economía, Industria y Competitividad, Gobierno de España | SAF2016-76565-R | Raül Andero Galí |
| Ministerio de Economía, Industria y Competitividad, Gobierno de España | SAF2017-87928-R | Angel Barco |

| Funder | Grant reference number | Author |
|---|---|---|
| Generalitat Valenciana | PROMETEO 2019/020 | Isabel Perez-Otaño |
| Ministerio de Economía, Industria y Competitividad, Gobierno de España | CSD2008-00005 | Isabel Perez-Otaño |
| Ministerio de Economía, Industria y Competitividad, Gobierno de España | SAF2013-48983R | Isabel Perez-Otaño |
| Ministerio de Economía, Industria y Competitividad, Gobierno de España | SAF2016-80895r | Isabel Perez-Otaño |
| National Institutes of Health | NS76637 | Steven J Tavalin |
| University of Tennessee | UTHSC College of Medicine Funds | Steven J Tavalin |
| Agence Nationale de la Recherche | GluBrain3A | Pierre Paoletti |
| Instituto de Salud Carlos III | AC19/00077 | Raül Andero Galí |
| Ministerio de Economía, Industria y Competitividad, Gobierno de España | RYC2014-15784 | Raül Andero Galí |
| Ministerio de Economía, Industria y Competitividad, Gobierno de España | SEV-2013-0317 | Isabel Perez-Otaño |
| Ministerio de Economía, Industria y Competitividad, Gobierno de España | SEV-2017-0723 | Isabel Perez-Otaño |
| Brain and Behavior Research Foundation | NARSAD Independent Investigator Award | Isabel Perez-Otaño |
| Ministerio de Economía, Industria y Competitividad, Gobierno de España | BFU-2016-80918-R | John F Wesseling |
| Agencia Estatal de Investigación | PID2019-111112RB-I00 | Isabel Perez-Otaño |

The funders had no role in study design, data collection and interpretation, or the decision to submit the work for publication.

#### Author contributions

María J Conde-Dusman, Partha N Dey, Conceptualization, Investigation, Methodology, Writing – original draft, Writing – review and editing; Óscar Elía-Zudaire, Investigation, Methodology, Writing – review and editing; Luis G Rabaneda, Conceptualization, Investigation, Methodology; Carmen García-Lira, Teddy Grand, Victor Briz, Eric R Velasco, Raül Andero, Sergio Niñerola, Angel Barco, Pierre Paoletti, Fabrizio Gardoni, Steven J Tavalin, Investigation; John F Wesseling, Conceptualization, Investigation, Writing – original draft, Writing – review and editing; Isabel Perez-Otaño, Conceptualization, Funding acquisition, Investigation, Methodology, Project administration, Writing – original draft, Writing – review and editing

#### Author ORCIDs

María J Conde-Dusman http://orcid.org/0000-0001-6841-2181
Óscar Elía-Zudaire http://orcid.org/0000-0002-1396-834X
Victor Briz http://orcid.org/0000-0001-6936-0918
Angel Barco http://orcid.org/0000-0002-0653-3751
Pierre Paoletti http://orcid.org/0000-0002-3681-4845
Fabrizio Gardoni http://orcid.org/0000-0003-4598-5563
Steven J Tavalin http://orcid.org/0000-0001-7169-0932
Isabel Perez-Otaño http://orcid.org/0000-0002-7222-8202

## Ethics

All procedures were conducted in accordance with the European and Spanish regulations (2010/63/UE; RD 53/2013) and were approved by the Ethical Committee of the Generalitat Valenciana (2017/VSC/PEA/00196). For the cued-fear conditioning experiments, ethic protocols were approved by the Committee of Ethics of the Universitat Autónoma de Barcelona and the Generalitat de Catalunya.

## Decision letter and Author response

Decision letter https://doi.org/10.7554/eLife.71575.sa1
Author response https://doi.org/10.7554/eLife.71575.sa2

---

# Additional files

## Supplementary files

• Transparent reporting form

## Data availability

RNAseq data have been deposited at GEO-NCBI under the access code GSE175920.

The following dataset was generated:

| Author(s) | Year | Dataset title | Dataset URL | Database and Identifier |
|---|---|---|---|---|
| Conde-Dusman MJ, Perez-Otaño I | 2021 | RNAseq data in cultured nerons | https://www.ncbi.nlm.nih.gov/geo/query/acc.cgi?acc=GSE175920 | NCBI Gene Expression Omnibus, GSE175920 |

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
