## [Decision Letter]

**Acceptance summary:**

This manuscript is of interest to those studying synaptic molecular mechanisms underlying learning. It identifies a key role for a particular class of developmentally regulated glutamate receptor in the activity-dependent regulation of dendritic protein synthesis. Multiple genetic, biochemical, molecular biological experiments strongly support the existence of the molecular mechanism proposed by the authors. Behavioral experiments with mutant mice suggest, but do not yet prove, that the mechanism contributes to memory formation in adult animals.

**Decision letter after peer review:**

Thank you for submitting your article "Control of protein synthesis and memory by GluN3A-NMDA receptors through inhibition of GIT1/mTORC1 assembly" for consideration by *eLife*. Your article has been reviewed by 3 peer reviewers, including Mary B Kennedy as Reviewing Editor and Reviewer #1, and the evaluation has been overseen by Gary Westbrook as the Senior Editor. The following individual involved in review of your submission has agreed to reveal their identity: Kobi Rosenblum (Reviewer #3).

The reviewers have discussed their reviews with one another, and the Reviewing Editor has drafted this letter to help you prepare a revised submission. The editors and reviewers were interested in the work, but had a number of concerns that we hope you can address. The list is long but we think attention to these issues would greatly enhance the impact of the work.

Essential revisions:*Reviewer #1:*

1. The presentation of the RNAseq data is confusing. It is clear over-all that the effects of GluN3A are not at the level of transcription; however, the labels indicating various messages in Figure 1D are not effective because it is not clear which dots are indicated. The authors should eliminate all labels except those for Arc and c-Fos, perhaps adding arrows. Figure 1D should be moved to one of the two figure 1 supplements.

2. In contrast, the data in Figure 2 supplement 1 are important for the overall conclusions. This figure could be added to the text as an additional figure. There are not limitations on the number of figures for an *eLife* paper.

3. Figure 4 supplement 1 should be added to Figure 4 as a quantitation of Figure 4 F.

4. The following statement from the Fiuza paper should be included in the methods section. "Proteins were transferred onto PVDF membranes (GE Health- care). After incubation with primary antibodies, membranes were incubated with secondary HRP-conjugated anti-mouse or anti-rabbit antibody (1:10,000; GE Healthcare). Films were scanned, and nonsaturated immunoreactive bands were quantified with ImageQuant 5.2 (Molecular Dynamics)." Readers should not have to search for another paper to understand the basic quantitative method used in several figures. Because the method is non-linear, it is important that readers know that only non-saturated bands were quantified.

5. Figure 5 Figure supplement 1 is mentioned in the text, but is not present in the supplementary figures. This should be fixed.

6. The limitations of interpretation of the i.p injection of rapamycin into grin3a knockout mice should be clarified. Because mTORC1 is important for many fundamental cell processes, the effect on memory could be non-specific. Thus, its effect on memory cannot be isolated to synaptic effects of rapamycin.

The Discussion contains some interesting ideas, but is difficult to read and needs clarification and reorganization. Several compound sentences should be divided or clarified.

7. In the last sentence of the third paragraph: change to … at specific developmental times and/or in specific subsets of synapses in adult brains.

8. In the first sentence of the third paragraph: change to … as shown by phosphorylation of GIT1-bound mTOR on Ser2448, an event that is stimulated by activation of NMDA receptors (Sutton and Chandler, 2002) and is amplified by feedback from the downstream target of mTORc1, S6Kinase (Chiang and Abraham, 2005).

9. Third paragraph line 6: Sentence should be: Fourth, association of GIT1 with GluN3A is required for mTORC1 modulation as demonstrated by the fact that expression in the grin3a knockout of a GluN3A mutant lacking the GIT1-binding site does not rescue the increased assembly of GIT1 with mTORC1(Figure 5) or the increased activation of synaptic mTORC1 (Figure 3).

10. First sentence of fourth paragraph: Sentence should be: … two central mechanisms in spines that are necessary for memory consolidation – actin cytoskeletal rearrangements and protein synthesis. (new sentence) This action would be analogous to the translational repression by FMRP/CYFIP1 complexes ….

11. Last sentence of fourth paragraph should be split: …addressed. It remains to be established whether GluN3A and GluN2B share common mechanisms.

12. In the second sentence of the fifth paragraph, it would be more accurate to replace the phrase "GO or NO GO biochemical environment" with "enhanced or repressed biochemical environment …"

13. The above sentence should be ended after "(figure D)" The next two sentences should be reversed as follows: "Synaptic GluN3A levels are down-regulated by sensory experience and can be controlled at the level of individual synapses by activity-dependent endocytosis (Perez-Otano et al., 2001). Thus, regulation of mTORC1 by GluN3A may also depend on the activity history of individual synapses, which is a key aspect in theories of memory consolidation (Redondo and Morris, 2011)."

14. The next sentence should be divided: "Removal of GluN3A-NMDARs from active synapses would drive formation of nearby GIT1/mTORC1 complexes. This would increase the potential for dendritic translation of activity-regulated mRNAs near active synapses, and might give active synapses an advantage for consolidation over less-active synapses. Hence, competition for active mTORC1 could provide a means for selective synapse stabilization and memory storage. Defects in mTORC1 regulation might permit consolidation of memories that would otherwise be lost.

15. Similarly, the next sentence should be divided: "Such a competition-based model is supported by the localization of GluN3A to subsets of adult synapses (Roberts et al., 2009). It is also supported by the observations that in Grin3a-/- mice, the level of GIT1/mTORC1 is increased and these animals exibit enhanced capacity for memory storage as shown by their performance in weak training protocols that are normally insufficient for stable memory formation in wild-type mice." Here the effect of rapamycin should not be mentioned because it could be non-specific.

16. The next sentence is incorrect. I believe it should say: "The restriction of dendritic translation to sites near active synapses underlies phenomena such as the competition between spines for lasting LTP expression (Fonseca …)" The meaning of the term "clustered dendritic plasticity" is unclear. Do you mean "potentiation of synapses in clusters along the dendrite"?

17. In the last sentence of this paragraph, the term "these phenomena" is vague. Be specific about which phenomena you mean.

18. In the second sentence of the fifth paragraph, the meaning is unclear because the construction is not parallel. It should read: "… and irrelevant events or associations are filtered out … "#

19. The sentence that begins: " The reversal by rapamycin should be modulated or eliminated altogether. "strongly suggests" should be changed to "is consistent with the notion"

20. The sentence beginning "As far as tested here … should begin a new paragraph. It begins a new topic.

21. In the sixth sentence of the next section: The phrase "lack neuronal/synapse specificity" is vague. Do you mean "are not specifically expressed only in neurons and synapses.

*Reviewer #3:*

General

22. One can say mRNA translation or protein synthesis but not translation. I believe that careful and articulate writing would help to improve the manuscript.

23. If they want to convey the message it's a protein synthesis dependent Long Term Memory we are talking about, I would emphasize the Fear Conditioning (1 hr versus 24 hr or more). The Conditioned Taste Aversion helps to show it's a border line effect that can be detected in weak but not strong paradigms.

24. Injecting rapamycin i.p. can do many things, it is not obvious to connect the genetic manipulation to mTOR dependent effect but their data clearly points at this direction.

Introduction.

25. It is believed- "Brains are made of complex neuronal networks" (the brain is made up of more than neurons) and "memories are encoded by modifying the synaptic connections between them" (There may also be non-synaptic modifications that contribute to memory storage).

26. "Encoding" requires de novo mRNA and protein synthesis in response to neuronal activity and sensory experience" (Do you mean consolidation? You may have short term memory.)

27. "but eIF2α affects general mRNA translation and evidence for a role in local translation is lacking (Sharma et al., 2020; Shrestha et al., 2020b)." As a matter of fact, there is high correlation between proteome of eIf2alpha and learning (its not general as the authors claim). Please look at Sharma et al., 2020.

28. Long and not clear sentence- "Negative regulation is mediated by inhibition of the assembly of mTORC1 complexes that contain the postsynaptic adaptor GIT1 (G protein-coupled receptor kinase-interacting protein) and Raptor, are located at or near synaptic sites, and couple mTORC1 kinase activity to synaptic stimulation. It would be better to break the three clauses up into sentences.

Results

29. Figure 1e representative, MG 132 seems to induce expression in Zif268 and arc as measured by WB?

30. Figure 6. "These results showed that GluN3A deletion facilitates spatial learning and memory without the unwanted effects associated to other modulators of translation." I do not understand what these sentence mean and what these results mean? there is no learning curve for the WT? if you do repeated measure after the reversal there might be a difference in the curve, what does it mean? There is no difference on latency after 7 days but statistical difference in the probe test? What does it mean?

31. "We then assessed long-term memory formation". 24 hrs. between trial in the Morris Water Maze (MWM) do not consider Long Term Memory?

32. Rapamycin erased the weak Conditioned Taste Aversion (CTA) memory in Grin3a-/- mice (Figure 6H), supporting the notion that disinhibited mTOR signaling causes the cognitive enhancement. Would it be the case if anisomycin or other Protein Synthesis Inhibitors will not have an effect?

33. Also Injecting an inhibitor i.p. (Rapamycin) in CTA is an issue since you inject twice and it may affect behavior in different ways.

34. As for the extinction, what is the p value for day1 Fear Extinction? This is also a memory test and if there are differences, it should be measured.

Discussion

Physiological relevance, Do you think GluN3A is a removable constraint on memory in the adult brain? Or is it a way to inhibit plasticity?

"for spine and memory consolidation" – there is no spine consolidation, what do you mean?

---

## [Author Response]

Essential revisions:Reviewer #1:1. The presentation of the RNAseq data is confusing. It is clear over-all that the effects of GluN3A are not at the level of transcription; however, the labels indicating various messages in Figure 1D are not effective because it is not clear which dots are indicated. The authors should eliminate all labels except those for Arc and c-Fos, perhaps adding arrows.

We have remodeled Figure 1D as suggested.

Figure 1D should be moved to one of the two figure 1 supplements.2. In contrast, the data in Figure 2 supplement 1 are important for the overall conclusions. This figure could be added to the text as an additional figure.

We agree with the reviewer and figure 2 supplement 1 is now part of the main manuscript.

There are not limitations on the number of figures for an eLife paper.3. Figure 4 supplement 1 should be added to Figure 4 as a quantitation of Figure 4 F.

Done as suggested.

6. The limitations of interpretation of the i.p injection of rapamycin into grin3a knockout mice should be clarified. Because mTORC1 is important for many fundamental cell processes, the effect on memory could be non-specific. Thus, its effect on memory cannot be isolated to synaptic effects of rapamycin.

We have moderated our claim as suggested. The original sentence: “The reversal by rapamycin strongly suggests” has been changed to “The reversal of rapamycin is consistent with the notion that the enhanced readiness of the mTORC1 translational machinery in GluN3A-deficient mice expands the range for consolidation of memory traces”. We have additionally expanded the Discussion to mention potential non-synaptic effects of rapamycin.

The Discussion contains some interesting ideas, but is difficult to read and needs clarification and reorganization. Several compound sentences should be divided or clarified.

We thank the reviewer for his careful reading and suggestions. We have included the changes indicated in points 6 to 20 to the Discussion section.

Reviewer #3:General22. One can say mRNA translation or protein synthesis but not translation. I believe that careful and articulate writing would help to improve the manuscript.

We agree with the reviewer and have corrected the instances where the incorrect term “protein translation” was used.

23. If they want to convey the message it's a protein synthesis dependent Long Term Memory we are talking about, I would emphasize the Fear Conditioning (1 hr versus 24 hr or more). The Conditioned Taste Aversion helps to show it's a border line effect that can be detected in weak but not strong paradigms.

We agree with the reviewer. In the behavioural section we now state that we used the two paradigms to assess associative memory formation, and emphasize how the fear conditioning allows to distinguish between unchanged short-term (1 h) and enhanced long-term (24h, 48h, 7 d) memory formation.

24. Injecting rapamycin i.p. can do many things, it is not obvious to connect the genetic manipulation to mTOR dependent effect but their data clearly points at this direction.

We have moderated our claim as suggested. The sentence that stated: “The reversal by rapamycin strongly suggests” has been changed to “The reversal by rapamycin is consistent with the notion that the enhanced readiness of memory traces”. We have additionally expanded the Discussion to address potential non-synaptic effects of rapamycin.

Introduction.25. It is believed- "Brains are made of complex neuronal networks" (the brain is made up of more than neurons) and "memories are encoded by modifying the synaptic connections between them" (There may also be non-synaptic modifications that contribute to memory storage).

We have rewritten our beginning sentence. It is widely assumed that memory involves synaptic modifications, but we agree with the reviewer that non-synaptic modifications such as changes in neuronal excitability or metabolic states likely contribute to memory storage. Our purpose was to focus the Introduction in the subject of our study and we did not mean to exclude other mechanisms.

26. "Encoding" requires de novo mRNA and protein synthesis in response to neuronal activity and sensory experience" (Do you mean consolidation? You may have short term memory.)

We have changed the term “encoding” for “lasting encoding”.

27. "but eIF2α affects general mRNA translation and evidence for a role in local translation is lacking (Sharma et al., 2020; Shrestha et al., 2020b)." As a matter of fact, there is high correlation between proteome of eIf2alpha and learning (its not general as the authors claim). Please look at Sharma et al., 2020.

We have re-written this sentence. The current view is that elF2α phosphorylation, by blocking the action of the initiation factor elF2B, reduces overall levels of protein synthesis. Yet we agree with the reviewer that it is unclear whether elF2α de- phosphorylation might drive transcript-specific translation in settings such as learning (for instance as a function of the abundance of constitutively available or activity-induced mRNAs). The reviewer is also correct that Sharma et al., in their excellent 2020 study attempted to address this issue by mapping the translatome and proteome of CA1 hippocampus in mice with reduced elF2α phosphorylation (see also Eacker et al., BioxRiv https://doi.org/10.1101/169425). Both studies identified different sets of translational changes that resembled those induced by learning, suggesting that the phosphor-status of elF2α might be a significant contributor following a memory experience.

28. Long and not clear sentence- "Negative regulation is mediated by inhibition of the assembly of mTORC1 complexes that contain the postsynaptic adaptor GIT1 (G protein-coupled receptor kinase-interacting protein) and Raptor, are located at or near synaptic sites, and couple mTORC1 kinase activity to synaptic stimulation. It would be better to break the three clauses up into sentences.

The sentence has been re-written as suggested.

Results29. Figure 1e representative, MG 132 seems to induce expression in Zif268 and arc as measured by WB?

The reviewer is correct. MG132 potentiates the BDNF induction of Arc and Zif268 by inhibiting the proteosomal degradation that contributes to terminate the IEG (compare BDNF alone vs BDNF + MG132) (see Rao and Finkbeiner, Nature Neuroscience 9,887, 2006; or Mabb et al., Neuron 82, 1299, 2014).

30. Figure 6. "These results showed that GluN3A deletion facilitates spatial learning and memory without the unwanted effects associated to other modulators of translation." I do not understand what these sentence mean and what these results mean? there is no learning curve for the WT? if you do repeated measure after the reversal there might be a difference in the curve, what does it mean? There is no difference on latency after 7 days but statistical difference in the probe test? What does it mean?

Two-trial per day Morris water maze experiments as the one in Figure 6 (now Figure 7; another example in females can be found in the Figure 7 – supplement figure 1B) often do not show a clear learning curve and latency averages are noisy when compared to the classical 4 trails per day (Figure 7 —figure supplements 1A). However, mice eventually learn the platform position as evident in the probe test conducted at day 7.

The sentence summarizing the first set of behavioral testing meant to emphasize differences between the behavioral outcome of GluN3A deletion with the observed after other manipulations of translation. For instance, GluN3A deletion leads to enhanced spatial memory in the Morris water maze without alterations in memory flexibility in the same test or signs of perseveration in the alternation test. By contrast: (i) reducing phosphorylation of elF2α in lateral amygdala enhances memory strength but reduces behavioral flexibility (Strestha et al., Nature Neuroscience 2020; Trinh et al., Cell Reports 2012); (ii) genetic removal of the translation modulator 4EBP impairs T-maze spontaneous alternation (Banko et al., Neurobiology of learning and memory 2007); (iii) enhancing mTOR signaling via brain disruption of FKBP12 enhances contextual fear memory but compromises behavioural flexibility in the Morris water maze and other tasks (Hoeffer et al., Neuron 2008). Our current plan is to complete the behavioural characterization of Grin3a KO mice and see if other domains of memory of cognition are compromised.

31. "We then assessed long-term memory formation". 24 hrs. between trial in the Morris Water Maze (MWM) do not consider Long Term Memory?

We agree with the reviewer that 24 hours is considered long-term memory. However, Morris water maze testing involves cumulative training with several trials per day and is not normally used to assess long-term memory. To avoid misunderstanding, we have changed the sentence to: “We then assessed associative memory formation”.

32. Rapamycin erased the weak Conditioned Taste Aversion (CTA) memory in Grin3a-/- mice (Figure 6H), supporting the notion that disinhibited mTOR signaling causes the cognitive enhancement. Would it be the case if anisomycin or other Protein Synthesis Inhibitors will not have an effect?

We did not perform this experiment, but we agree that if would support our claim and we will include in our next round of experiments as Grin3a KO mice become available.

33. Also Injecting an inhibitor i.p. (Rapamycin) in CTA is an issue since you inject twice and it may affect behavior in different ways.

We agree, and have moderated our claims as suggested (see response to point 30).

34. As for the extinction, what is the p value for day1 Fear Extinction? This is also a memory test and if there are differences, it should be measured.

For the results reported in the manuscript, we used a rmANOVA using mean freezing during each FE session (Fes) as within-subject factor and group (Grin3a -/- Wild-type) as between-subject factor. Here, we found an effect of fear extinction sessions, meaning that animals were successfully extinguishing fear throughout time (F(3,75)=84.035, p<0.001), but without differences between groups (tx*Fes) (F(3,75)=1.402, p=0.254). A between-subject (F(1,25)=4.402, p=0.046) showed that Grin3a -/- animals had overall lower freezing rates compared to Wild-type animals.

As the interaction tx*Fes was not significant, we proceeded to analyze freezing in each FE session individually. Freezing levels were not different among treatments in FE1 (t(25)=0.760, p=0.455) or FE2 (t(25)=1.948, p=0.063), but Grin3a -/- mice had significantly less freezing compared to Wild-type in FE3 (t(25)-2.127, p=0.043) and FE4 (t(25)=2.108, p=0.045). The information on FE day 1 requested by the reviewer has now been included in the figure legend.

Given your observation, we considered important to carry out additional analysis to explore if there were memory differences within each FE session. This would inform if the differences were arising from FE session performance or memory consolidation throughout the days. We did this by averaging freezing during 5 CS presentations for each FE session (CS 1 to 5, CS 6 to 10, CS 11 to 15) and used them as within-subject factors in rmANOVA analysis. Within-session freezing levels were similar among groups in FE1 (F(2,50)=1.218, p=0.304), FE2 (F(2,50)=0.120, p=0.887), FE3 (F(2,50)=0.645, p0.529) and (FE4 (2,50)=0.988,p=0.376), meaning that animals extinguished fear at similar rates in each FE session independently of their genotype.

DiscussionPhysiological relevance, Do you think GluN3A is a removable constraint on memory in the adult brain? Or is it a way to inhibit plasticity?

1) As we mention in the Discussion, GluN3A is developmentally regulated but also retained in specific brain areas and subsets of synapses. Furthermore, neurons are endowed with specific mechanisms for local, activity-dependant removal from synapses. These properties make GluN3A a suitable candidate for a removable memory constraint.

2) We have considered the alternative possibility, that GluN3A synapses would be a conduit for short-term memories while GluN3A-lacking synapses would encode long-lasting memories. Discerning between these and other possibilities will require further work.

"for spine and memory consolidation" – there is no spine consolidation, what do you mean?

We have changed the wording of this sentence.